# Diurnal and Seasonal Variations of Dust Transport around the Tibetan Plateau: Insights from Multi-Source Observations

Xiaofeng Xu<sup>1</sup>, Zixu Xiong<sup>2</sup>, Jianming Gong<sup>1</sup>, Huilin Zhang<sup>1</sup>, Tianliang Zhao<sup>1</sup>, Qing He<sup>3</sup>

<sup>1</sup>Key Laboratory for Aerosol-Cloud-Precipitation of China Meteorological Administration, School of Atmospheric Physics, Nanjing University of Information Science and Technology, Nanjing 210044, China

<sup>2</sup>Zhenjiang Meteorological Bureau, Zhenjiang 212003, China

<sup>3</sup>Institute of Desert Meteorology, China Meteorological Administration, Urumqi 830002, China

Correspondence to: Xiaofeng Xu (xxf@nuist.edu.cn)

Abstract. Dust transport around the Tibetan Plateau (TP) plays a key role in regional climate and air quality, yet its seasonal and diurnal variability remains insufficiently understood. Here, we presented the spatiotemporal characteristics of dust transport in the TP region combining satellite observations, reanalysis datasets and ground-based measurements. A new method for dust mass concentration was developed and showed strong consistency with multiple products in both spatial and temporal scales. Our results revealed persistent dust transport belts on both the northern and southern sides of TP, with peaks in spring, the amounts of dust flux transported to TP were estimated for different directional sources, seasons and heights. The vertical variation and amount of dust backflow in Taklamakan Desert were analyzed. The diurnal characteristics of vertical resolved dust flux were presented in three-hour interval and four sections around TP. Overall, this study deepened the understanding of the dust climatology over the TP region from a satellite perspective.

#### 1 Introduction

The Tibetan Plateau (TP), known as the "Roof of the World", is the highest and largest plateau on Earth. It serves as the source of several major rivers in Asia, including the Yangtze, Yellow and Indus Rivers. The plateau's mechanical and thermal forcing has a profound influence on atmospheric circulation, monsoon precipitation, the transport of atmospheric materials, and airsea interactions (Duan et al., 2012; Huang et al., 2023; Liu et al., 2020; Wu, 2020), playing a critical role in both regional and global climate systems. In addition, TP is located near the Taklamakan Desert (TD), the Thar Desert (TH), and the Gobi Desert (GD), all of which are key sources of dust aerosols in Asia and over the world. The dust aerosols in this region (Xu et al., 2015) can cause an instantaneous atmospheric heating of up to 5.5 K/day, and significantly impact the atmospheric thermal and dynamic structure (Jia et al., 2018). This can even lead to the intensification of the Tibetan anticyclone and the northward shift and the strengthening of the Mei-Yu rain belt (Lau and Kim, 2018). The dust deposited on snow can reduce the surface albedo by accelerating the snow melting(Zhao et al., 2022a), which in turn has profound effects on ecosystems, water resources, and climate change (Wang et al., 2015).

30

45

Given the significant impact of TP and dust aerosols in the region, extensive research has been conducted on the sources, distribution, and transport of dust aerosols. In terms of model simulation, Liu et al. (2015) utilized the Spectral Radiation-Transport Model for Aerosol Species in conjunction with a non-hydrostatic regional model, revealing that dust plumes at altitudes of 7-8 km over the TP during summer mainly originated from TD. Due to topographical constraints, the northwesterly winds shifted to northeasterly, carrying the dust to the northern slope of the TP. Similarly, Chen et al. (2013) employed the WRF-Chem model to analyze a severe dust storm event in TD, finding that dust penetrated the boundary layer and extended into the upper troposphere of the northern TP, with dust transport flux of  $6.6 \, Gg \cdot day^{-1}$ . In a more comprehensive study, Hu et al. (2020) utilized the WRF-Chem model with tracer-tagging technique to track dust sources at different altitudes over the TP. Their findings showed that East Asian dust, mainly from the TD and GD, entered TP from the northern slope, with a mass flux of  $7.9 \, Tg \cdot yr^{-1}$ . In contrast, dust from North Africa and the Middle Eastern deserts entered TP from the west, with mass fluxes of  $7.8 \, and \, 26.6 \, Tg \cdot yr^{-1}$ , respectively. However, Zhang et al. (2024) highlighted a significant limitation in traditional WRF-Chem simulations, which often underestimated dust emissions over the TP. By adjusting the erodibility factor in the model, they estimated total dust emissions from the TP in 2018 to be  $258.82 \, Tg \cdot yr^{-1}$ . Despite these adjustments, it is important to note that the dust mass concentration (DMC) and transport flux simulated by models might vary considerably due to differences in model configurations and parameterization schemes, leading to substantial uncertainty (Zhao et al., 2020).

On the observational side, Jia et al. (2015) used backward trajectories and weather system analysis to show that summer dust over the TP primarily originated from TD, TH, and the Gurbantunggut Desert, with dust activity being closed linked to cold air advection and low-pressure systems. Using MISR and CALIPSO (Cloud-Aerosol Lidar and Infrared Pathfinder Satellite Operations) data, Xu et al. (2015) illustrated the distribution of aerosol optical thickness over the TP, noting that due to the elevation differences between the northern and southern slopes, aerosols were more likely to reach the TP via the northern slope. Xu et al. (2018) further reported that the dust mass flux was 1010 g across a 2° latitude band downstream of the TP in the upper troposphere in spring based on CALIPSO and ERA-5 datasets. They also found a distinct dust belt extending across the Pacific to North America, observable at altitudes above 6 km along the TP's downwind direction. In a more recent study, Han et al. (2022a) proposed an effective method of converting satellite-observed aerosol data into mass concentration and transport flux, and estimated the dust transport contributions from East Asian and South Asian deserts to mainland China and its neighboring seas based on CALIPSO observations, which were 214.28 and 30.43 Tg, respectively. However, despite these important findings, there remain limitations in observational studies concerning the three-dimensional distribution of DMC in this region. Furthermore, due to the limited spatiotemporal resolution of traditional satellites such as CALIPSO, estimating of diurnal variations in dust transport flux from surrounding deserts (TD, TH, GD) to the TP are still scarce. The operation of the CATS (Cloud-Aerosol Transport System) lidar offers a promising opportunity to fill this gap. While it does not allow continuous monitoring of a specific area throughout the entire day, it enables the reconstruction of an almost complete daily cycle for the region by statistically integrating CATS data over approximately 60 days (Chepfer et al., 2019; Yorks et al., 2016). This study thus applied the method of Han et al. (2022a) to derive relatively accurate seasonal and diurnal cycles of dust over TP and its surroundings based on CATS observational data. In the following, we describe the methods and data in

Section 2 and present the major results in Section 3, which include spatial changes, diurnal and seasonal variability, and vertical distributions of dust transport. Then, we conclude the study in Section 4.

# 2 Data and methods

70

As shown in Fig. 1, the main study area included TP and three surrounding deserts: the TD, GD, and TH. Various meteorological and aerosol products, including ground-based, satellite-observed and reanalysis datasets, were used in this study. Details of calculated parameters, source datasets and corresponding variables were shown in Table 1.

**Figure 1.** The topography of the study aera (23-45°N, 68-115°E). S1 to S4 were the selected boundaries for the TP-ward dust flux calculation and X1 to X2 were used to quantify the dust contribution of the Qaidam Basin to downstream regions.

# 75 **Table 1.** Data information.

| Calculated parameters                              | Datasets | Variables                                                                     |
|----------------------------------------------------|----------|-------------------------------------------------------------------------------|
| $\sigma_D$ (Dust extinction coefficient) at 532 nm | CATS     | Extinction_Coefficient_1064_Fore_FOV                                          |
|                                                    | CALIPSO  | Extinction_Coefficient_1064; Extinction_Coefficient_532                       |
| DMC (dust mass                                     | DustCOMM | MEE_mean (Dust_3D_MEE_seasonal.nc)                                            |
| concentration, $g \cdot m^{-3}$ ) and DMCC (dust   | MERRA-2  | DUSMASS (Dust Surface Mass Concentration); DUCMASS (Dust Column Mass Density) |
| column mass concentration, $g$ .                   | DustCOMM | Load_mean (Dust_Load_seasonal.nc)                                             |
| $m^{-2}$ )                                         | CHAP     | $PM_{10}$                                                                     |

90

|                                                               | CNEMC            | PM <sub>2.5</sub> ; PM <sub>10</sub>                                                                                          |  |
|---------------------------------------------------------------|------------------|-------------------------------------------------------------------------------------------------------------------------------|--|
| DFR (dust flux rate, $g \cdot m^{-1} \cdot s^{-1}$ )          | ERA5             | U-component of wind; V-component of wind; Vertical velocity                                                                   |  |
|                                                               | MERRA-2          | DUFLUXU (Dust column u-wind mass flux);                                                                                       |  |
|                                                               |                  | DUFLUXV (Dust column v-wind mass flux)                                                                                        |  |
| E ( dust exposure, $g$ ·                                      | GPWv4            | gpw_v4_population_count_rev11_2020_15_min.tif                                                                                 |  |
| $m^{-2}$ )                                                    | MERRA-2          | T2MAX, T2MIN                                                                                                                  |  |
| $g \cdot m^{-1} \cdot s^{-1}$ )  E ( dust exposure, $g \cdot$ | MERRA-2<br>GPWv4 | DUFLUXU (Dust column u-wind mass flux); DUFLUXV (Dust column v-wind mass flux)  gpw_v4_population_count_rev11_2020_15_min.tif |  |

# 2.1 Dust extinction coefficient profile

The CATS lidar was installed on the International Space Station (ISS), where it monitored the vertical distribution of atmospheric aerosols and clouds at 1064 nm from March 2015 to October 2017 (McGill et al., 2015). The ISS's orbit at a 51.6° inclination provided CATS a higher temporal resolution from the tropics to mid-latitudes than classical spaceborne sensors such as CALIOP (Cloud-Aerosol Lidar with Orthogonal Polarization). We performed quality control and calculations on the CATS product of Level 2 Operational (L2O) version 3.01 using the method described by Xiong et al. (2023). Two types of dust extinction coefficient  $\sigma_D$  at 1064nm were obtained: one is to average out the dust loading over an entire year, characterizing the average dust distribution in climatological state ( $\sigma_{D_A}$ ), the other considered the dust conditions only during dusty events, describing the short-term and diurnal changes of dust in dusty days ( $\sigma_{D_D}$ ). Due to the presence of airborne dust layers in the study aera,  $\sigma_{D_A}$  and  $\sigma_{D_D}$  didn't follow a strict exponential decay with height. Therefore, we relied solely on the available retrieved values for the calculations rather than interpolating at missing values. In most studies, the aerosol optical properties are described in the wavelength of 550nm, so we firstly introduce a method of converting the  $\sigma_D$  from 1064nm to 532nm by the aid of CALIPSO.

The CALIOP located on the CALIPSO satellite was widely used to study the distribution and transportation of aerosols since its launch in 2006 (Winker et al., 2013). CALIPSO v4.51 L2 aerosol profile product offered the aerosol extinction coefficient at both 1064 nm and 532 nm. The dust was picked by constraining the Atmospheric Volume Description, which described different feature types and characteristic subtypes. By extracting the CALIPSO  $\sigma_D$  profiles at two wavelengths on various altitudes in the study area from March 2016 to February 2017, the median profiles of the conversion ratio of extinction coefficient from 1064 nm to 532 nm ( $ratio = \sigma_{D_{-}532}/\sigma_{D_{-}1064}$ ) in different seasons were showed in Fig. 2. Based on the conversion ratio derived from the CALIOP products, the CATS  $\sigma_D$  profile at 532 nm was calculated from the CATS product at 1064 nm. Subsequently, these profiles were gridded to a horizontal resolution of 0.5°×0.5° and a vertical resolution of 60 m.

120

Figure 2. The conversion ratio ( $ratio = \sigma_{D.532}/\sigma_{D.1064}$ , left) derived from CALIPSO and the data volume of dust samples (right) used in calculation at different altitudes during the period from March 2016 to February 2017.

#### 2.2 Calculation of DMC and flux

MEE (m²·g⁻¹), which was defined as the total light extinction per unit mass of aerosol (Hand and Malm, 2007; Yu et al., 2012) was used to convert the σ<sub>D</sub> to dust mass concentration (DMC) in this study. The MEE was derived from the DustCOMM (Dust Constraints from joint Observational-Modelling-experiMental analysis) dataset (Adebiyi et al., 2020). This dataset integrated the global results of models, observations, and experiments on dust characteristics, providing an unprecedented accuracy of 550 nm dust MEE dataset in three-dimensions (3D) with the spatial resolution of 0.5° × 0.625° and 35 pressure
levels (Adebiyi et al., 2020). By combining globally averaged dust particle size distribution with spatial variability from model simulations, DustCOMM achieved a more precise dust size distribution in the 0.2–20 μm range, particularly reducing the underestimation of coarse-mode particles in previous models. Additionally, it represented dust particles as a collection of triaxial ellipsoids rather than simple spheres to minimize errors due to particle shape in MEE estimation. The resulting MEE had an error margin reduced to approximately 1%, rendering it nearly negligible compared to observational data (Adebiyi et al., 2020). The DMC (g · m⁻³) could be derived from the DustCOMM MEE and CATS σ<sub>D</sub> as follows:

$$DMC = \frac{\sigma_D}{_{MEE}},\tag{1}$$

Here, the resulting DMCs in the climatological state and in dusty days were calculated as  $\sigma_{D_-A}$  and  $\sigma_{D_-D}$ , respectively. Assuming the well mixing of dust in the lowest atmosphere, the average DMC below 180 m was considered as the surface dust mass concentration (DMCS). The total dust loading, also called dust mass column concentration (DMCC,  $g \cdot m^{-2}$ ), was the integral of DMCs from bottom to top of atmosphere. Then, the dust flux rate (DFR,  $g \cdot m^{-2} \cdot s^{-1}$ ) in three directions was calculated as follows:

130

$$DFR_{-}U = DMC \cdot u, \qquad (2)$$

$$DFR_{-}V = DMC \cdot v$$
, (3)

$$DFR_{-}W = DMC \cdot w , \qquad (4)$$

Where, u, v and w represented the wind speeds in the north-south, east-west, and vertical directions derived from ERA5 reanalysis dataset, respectively. The dust column flux rates of DFRC\_U and DFRC\_V  $(g \cdot m^{-1} \cdot s^{-1})$  were the columnal integrals of the corresponding variables. And total DFRC was the composite of DFRC U and DFRC V.

To estimate the contribution of dusts from the Qaidam Basin to downstream cities, it was necessary to exclude the influence from other dust sources, such as TD. As shown in Fig. 1, sections X1 and X2 were defined at the western entrance and eastern exit of the Qaidam Basin, respectively. According to statistics, the annual precipitation in the Qaidam Basin was only 88.31 mm during the period from 1981 to 2010 (Wang et al., 2014). The region experienced an arid climate due to the influence of westerly winds and the Mongolian high-pressure system. Consequently, only the dry deposition process within in the basin was considered, while the wet deposition process was negligible. Then, the net dust flux ( $F_{net}$ ) was calculated following the algorithm proposed by Han et al. (2022a), as below:

35 
$$f_D = \frac{F_D}{F_1} = \frac{1}{F_1} \sum_i DMCS_i \cdot V_D \cdot S_i$$
, (5)

$$F_1^k = F_1 \cdot (1 - f_D)^k \ (k = 0, 1, ..., n) \ , \tag{6}$$

$$F_2^* = F_2 - F_1^n \,, \tag{7}$$

$$F_{net} = \frac{F_2^*}{I},\tag{8}$$

where  $F_1$  and  $F_2$  represented the total integrated dust fluxes at the X1 and X2 cross-sections, respectively.  $DMCS_i$  and  $S_i$  denoted the surface dust concentration and grid area at the i-th grid point since X1.  $V_D$  was the dry deposition rate with a constant of  $0.02 \ m \cdot s^{-1}$  (Han et al., 2022a; Hsu et al., 2009; Wang et al., 2012).  $f_D$  represented the dust lost fraction at X1 caused by dust deposition and was assumed to be a constant in the basin. Therefore,  $F_1^k$  denoted the remaining integrated dust flux of  $F_1$  after travelling k grids eastward from X1.  $F_1^n$  was the remaining dust flux at X2 in the Qaidam Basin.  $F_2^*$  was the total net dust flux at X2 and L was the length of the X2.  $F_{net}$  was the net flux per unit length along X2 section.

It was well known that dust could carry various bio-particles and microorganisms, potentially posing health risks to human health along its trajectory (Gonzalez-Martin et al., 2014; Zhang et al., 2016). Furthermore, exposure to non-optimal temperatures could heighten the incidence and mortality risks of sensitivity-related diseases triggered by particulate matter (Wang et al., 2022; Zhang et al., 2018). In order to simply evaluate the impact of dust transport on human societies, we defined a factor of dust exposure (E,  $g \cdot m^{-2}$ ) as follows:

$$E = \frac{P_{i,j} - P_{min}}{P_{max} - P_{min}} \cdot f(T) \cdot DMCC_{i,j} , \qquad (9)$$




where  $P_{i,j}$  and  $DMCC_{i,j}$  represented the population density and column dust mass concentration at each grid point, while  $P_{max}$  and  $P_{min}$  referred to the maximum and minimum of grid population density in the study area, respectively. The function f(T) served as the temperature adjustment factor, as illustrated in Fig. 3. Its values were approximately determined based on the relative risk associated with non-optimal versus optimal temperatures (Wang et al., 2022). Specifically, when f(T) = 1, temperature had no modifying effect on the health impact of dust exposure. Conversely, higher f(T) values indicate stronger adverse effect of dust on human. To better capture the influence of extreme temperatures on health, the daily minimum (maximum) 2-meter air temperature from MERRA-2 was used for the winter (summer) half year. The population data was from the Gridded Population of the World, Version 4.11 (GPWv4), developed by the Center for International Earth Science Information Network (CIESIN) at Columbia University. The dust exposure aimed to qualify the extent to which people were affected by dust in different regions based on actual observations rather than numerical simulations, thereby enhancing our understanding of the potential effects of dust on human health. Based on the spatial distribution of dust exposure, a threshold of 5  $mg \cdot m^{-2}$  was adopted to define the high-exposure regions.

**Figure 3.** The temperature adjustment function f(T) from Wang et al. (2022).

# 2.3 Other auxiliary data

To assess the reliability of the results, several auxiliary datasets were used. Monthly DMCC, DMCS and DFR from MERRA-2 reanalysis (MERRA2\_400.tavgM\_2d\_aer\_Nx) with a horizontal resolution of  $0.5^{\circ} \times 0.625^{\circ}$  were used to compare with our results from CATS. The ground-based observations of PM<sub>10</sub> concentration were from the China National Environmental Monitoring Center (CNEMC). Besides, the CHAP dataset, which was produced by an advanced machine learning method,

offers a high-resolution (1 km)  $PM_{10}$  data in China, with a high correlation and a low RMSE compared with station observation (Wei et al., 2021). The  $PM_{10}$  concentrations from CHAP and CNEMC were used to evaluate our calculations of DMCS from CATS. The wind fields of 2016 were derived from the hourly and monthly ERA5 reanalysis products, with a horizontal resolution of  $0.5^{\circ} \times 0.5^{\circ}$  and 20 pressure levels ranging from 1000 to 300 hPa.

#### 3 Results






#### 3.1 Dust transport features from four datasets

Fig. 4 showed the seasonal spatial distribution of climatic-state dust column mass concentration (DMCC) and dust surface mass concentration (DMCS) from four datasets of CATS, MERRA-2, DustCOMM and CHAP during 2015-2017. As to DMCC, CATS demonstrated a good consistency with the other two datasets in both spatial and temporal distributions. In the study aera, the DMCC in the three main dust sources (TD, TH and GD) was significantly higher than that in the other regions throughout the whole year, with the highest loading in TD, which is consistent with the previous studies (Li et al., 2023; Xu et al., 2015, 2020). A long-standing dust belt existed over TD and surroundings, which could convey dust southward across the Qaidam Basin to TP, and eastward to affect the air quality of East China and North China. It presented consistently a larger dust loading of 2 to 3 times in spring (MAM) and summer (JJA) than in autumn (SON) and winter (DJF) in all three datasets. The differences of annual average among these datasets were also acceptable, showing annual regional means of 0.111 (CATS), 0.138 (MERRA-2) and 0.122  $g \cdot m^{-2}$  (DustCOMM), respectively. The mean DMCCs of MERRA-2 and DustCOMM were about 24% and 9.9% higher than the CATS value, respectively. The previous studies also noted the overestimation of MERRA-2 in deserts and surrounding regions (e.g., East Asia deserts, Northwest China, and Australia) relative to CALIPSO and AERONET (Han et al., 2022a; Mukkavilli et al., 2019; Sun et al., 2019). This discrepancy may be attributed to limitations in the chemical transport model used by MERRA-2 (Gueymard and Yang, 2020). In describing the seasonal variations of DMCC, both CATS and DustCOMM showed that the highest DMCC appeared in TD during spring, while that of MERRA-2 peaked in summer. This suggested that MERRA-2 might overestimate dust loading of TD more prominently in summer. Additionally, compared to CATS and DustCOMM, MERRA-2 significantly underestimated dust concentrations in the high dust loading regions in TD during spring. Shi et al. (2023) found that MERRA-2's simulation accuracy in China significantly decreased when the total AOD exceeded 1, which occurred frequently due to the dust events in spring. In brief, CATS provided advantageous results, offering higher temporal and spatial resolution, which facilitated the analysis of diurnal variations of dust.


Figure 4. The seasonal distributions of (A) DMCC and (B) DMCS from different datasets.

The distribution characteristics of DMCS was similar to DMCC. Regions of high-value DMCS were found along the southern edge of TD, the southwestern part of TH, the northern part of the GD, and the Qaidam Basin, which corresponded well to the aeras with high wind speeds (Fig. 5). The regional annual average DMCSs of CATS and MERRA-2 were 75 and  $66.1 \ \mu g \cdot m^{-3}$ , respectively. The CATS DMCS was slightly higher than MERRA-2 result. This was because the CATS retrievals had more missing values over the areas with weaker dust concentrations on the TP, which might lead to an overestimation of the overall results. Since the primary particulate type in this region was dust, CHAP PM<sub>10</sub> could partially



reflected the DMCS characteristics. Compared to MERRA-2, CATS and CHAP datasets showed greater consistency in depicting the distribution of DMCS and provided more coherent representations of high value regions within TD. Both datasets identified DMCS maximum over the southwestern part of TD in spring, whereas MERRA-2 located it in the northeast. Similarly, the spatial patterns and magnitudes of DMCC derived from CATS were more consistent with those from DustCOMM, with finer details in the distribution. In contrast, MERRA-2 exhibited notable discrepancies. For instance, MERRA-2 showed a summer peak in DMCC over TD, whereas both CATS and DustCOMM indicated a maximum in spring. Furthermore, MERRA-2 maintained a relatively uniform spatial distribution pattern across seasons for both DMCC and DMCS, with seasonal variations primarily reflected in the magnitude rather than the spatial structure. Additionally, we matched data from over 170 CNEMC sites within the study area with the CATS data. Considering the impact of fine-mode particles, we used the difference of surface observed PM<sub>10</sub> and PM<sub>2.5</sub> as the surface dust concentration, and then calculated the seasonal mean values and correlations between them. We also calculated the averages and correlations between CATS and MERRA-2 at matching grids in different seasons. The results showed that CATS's DMCS had an acceptable correlation with both ground-based observations and reanalysis data (Table 2). The correlation coefficients ranged from 0.4 to 0.6, with the highest correlation in spring. In conclusion, the dust concentrations retrieved from CATS effectively illustrated the distribution of dust in the study region and exhibited a reliable agreement with reanalysis data and surface observations.



Figure 5. Seasonal mean wind fields at 10 m and on three pressure levels of 850, 700, and 500 hPa from the ERA5 reanalysis dataset.

**Table 2.** The seasonal means, correlation coefficients (R) and sample sizes of DMCS.

|               | Season | Mean          | R    | Samples |
|---------------|--------|---------------|------|---------|
|               | MAM    | 50.46, 68.91  | 0.64 | 173     |
| CNEMC vs CATS | JJA    | 31.83, 19.48  | 0.46 | 166     |
| CNEMC VS CATS | SON    | 39.18, 26.46  | 0.45 | 170     |
|               | DJF    | 48.64, 35.57  | 0.40 | 171     |
|               | MAM    | 95.70, 120.46 | 0.58 | 3010    |
| MERRA-2 vs    | JJA    | 91.42, 79.33  | 0.51 | 2509    |
| CATS          | SON    | 79.18, 49.66  | 0.58 | 2979    |
|               | DJF    | 57.33, 50.40  | 0.41 | 2745    |

# 230 3.2 Seasonal characteristics of dust transport

To elucidate the dust transport features in different directions within the study area, Fig. 6 illustrated the distributions of DFRC U (column integral dust flux rates in the east-west direction), DFRC V (column integral dust flux rates in the northsouth direction), and total DFRC derived from MERRA2 and CATS datasets, separately. DFRC represented the net dust flux within the atmospheric column. Fig. 7 showed the vertical distribution of dust transport in several cross-sections within TD and TP. The positive values of DFRC U and DFRC V corresponded to eastward and northward, respectively. Two dust transport belts could be found on northern and southern sides of TP, respectively. The northern belt extended from TD into the northeastern part of China, while the southern belt started from TH and spread northeast to the areas of 100° E. Both the magnitude and spatial extent of dust flux peaked in spring, with DFRC values exceeding  $100 \text{ kg} \cdot \text{m}^{-1} \cdot \text{day}^{-1}$ . Significant dust-carrying upward airflows were observed in the 850-600hPa layers over TD, TH and GD during spring and summer (Fig. 8). Additionally, the DFRC in Qaidam Basin was consistently greater than  $60 kg \cdot m^{-1} \cdot day^{-1}$  throughout the year, which proved it a stable and important channel transporting dust to the TP. Although GD exhibited a low dust uplift efficiency (Chen et al., 2017) and a relatively lower DMCC (Fig. 4), its DFRC was much higher than that of TD. This was attributable to the fact that, unlike TD, which was enclosed by mountainous terrain on three sides, GD possessed a more open topography. Deep convective mixing and frequent intrusions of cold air resulted in higher low-level wind speed in GD (Fig. 5), making it the largest dust contributor in East Asia (Chen et al., 2017; Zhang et al., 2008). Although the DMCC was still high in the three deserts during summer (Fig. 4), the transport flux and eastward extension of dust belt were markedly reduced. This may be due to the lowest wind speed in summer on both southern and northern sides of TP in levels of 500-700 hPa (Fig. 5). Additionally, surface wind in GD and the eastern TH shifted from northwesterly in spring to southeasterly in summer, which




partly hindered the transport of dust from the desert regions to the TP and the downstream regions. In autumn and winter, although the magnitude of dust transport decreased, the dust belt on the northern TP still could extend to the far eastern edge of the study area. The influx of strong northerly winds in winter enhanced the westerly jet stream, expanding the area of strong winds region (> $16m \cdot s^{-1}$ ) at 500hPa (Fig. 5p). As a result, large areas in the northwestern TP and downstream regions, extending as south as ~ $30^{\circ}$  N, were impacted by the northern dust belt.

**Figure 6.** Seasonal distribution of CATS DFRC (a-d), and DFRC\_U and DFRC\_V from CATS (e-l) and MERRA-2 (m-t) datasets, respectively. (Positive values of DFRC\_U and DFRC\_V corresponded to eastward and northward transport, respectively.)

Based on the distributions of DFRC\_U and DFRC\_V (Fig. 7), the dust entering TP has multiple sources, such as TD, TH, Central Asia, and even Africa (Hu et al., 2020; Mao et al., 2019). Notably, it existed a dust backflow throughout the year in TD (Fig. 7b). The dust swept from the eastern entrance into the interior of TD, which was especially apparent in spring and summer. Due to the obstruction of the Pamir Plateau and the Tianshan Mountains, the cold air mainly invaded from the east side. During spring and summer, intense surface-atmosphere heat exchange leaded to the formation of a thermal low pressure in TD (Han et al., 2005). The combined effect of the cold air and thermal low pressure caused the strong easterly and northeasterly winds at the TD surface, and subsequently led to the dust backflow. Additionally, two pathways of transporting dust from TD to TP were clear according to Fig. 6(i-l, q-t). The first path is of climbing the northern slopes of the Kunlun Mountains under the effect of rising air (Fig. 7a). The second path is of crossing the Qaidam Basin along with the northwesterly and northerly winds. In summer, the reduction of winds in 500hPa and the northward shift of the westerly jet stream to the


northern TP (Chen et al., 2017) diminished the eastward transport of the TD dust (Fig. 6f, n) while enhanced its southward transmission into the TP (Fig. 6j, r). Under the westerly and northerly winds, the GD dust was primarily transported eastward and southward to the downstream regions, and influenced less to the northeastern part of TP. TH is another significant dust source contributing to TP. The strong upward motion driven by the secondary circulation in this area provided essential dynamics for the vertical lifting of dust (Wang et al., 2021). Along with the increasing altitudes, the wind direction shifted from north to south, which facilitated the transport of floating dust to TP (Fig. 7). Additionally, a certain part of dust reaching TP was mainly from its eastern side of about 88° E. During summer, the southwest monsoon also provided significant assistance in lifting the dust and other aerosols onto the plateau (Liu et al., 2015).

**Figure 7.** Vertical cross-sections of (a) DFR\_V and (b) DFR\_U in TD and TP in spring. (The blue area near 38.5°N in (b) represents the easterly dust backflow.)



As shown in Fig. 8 significant dust-carrying upward airflows were observed in the 850-600 hPa layers over the three major dust source regions during spring and summer. The largest upward dust flux was in TD. Previous studies had revealed that airflows typically converged and ascended along the southern and northwestern margins of the Tarim Basin. This promoted the dust emission, and made these regions having the most frequent dust events in TD (Han et al., 2005; Yang et al., 2016). In TH, it showed the largest aera of dust uplift in summer. The highest surface temperature in summer intensified the convective instability in desert, leading to stronger convective activities. Meanwhile, the strong westerly winds above 700 hPa could further carry dust to more distant regions in the east, such as Yunnan Province in China and some parts of Southeast Asia. Additionally, a relatively large quantity of DFR\_W existed through the whole year at 500hPa over TP, which was an important dust source for the downstream areas.

**Figure 8.** Seasonal distribution of DFR\_W (dust flux rates at vertical direction) at four pressure levels from CATS. (The arrows represent the horizontal wind field. The positive values of DFR W corresponded to upward transport.)

#### 3.3 Dust backflow in TD

As mentioned above, given the west-to-east dust transport dominated in the study area, the year-round east-to-west dust backflow in the TD was notably unusual, and the vertical characteristics of dust transport deserved further investigation. Fig.





9 illustrated the zonal mean seasonal dust transport fluxes in TD. The dust backflows (east-to-west transport) primarily occurred in the lower atmosphere with the altitude below 2.5 km, which was slightly higher in spring and summer. This phenomenon, named "East irrigation" weather, is caused by the cold air that crosses the Tianshan Mountains and then enters the Tarim Basin. Besides, Han et al. (2022b) found that the nocturnal low-level jet during summer could also contribute to the westward dust transport. The amount of backflow dust exhibited distinct seasonal variations, which was maximum in spring with an average of 83.1  $ton \cdot km^{-2} \cdot day^{-1}$  (Fig. 9b). The backflow dust flux decreased with altitudes, showing the maximum of 134  $ton \cdot km^{-2} \cdot day^{-1}$  near the surface in spring (Fig. 9a). The dust backflow decreased a lot in autumn and summer, and the weakest was in winter, showing an average of 7.4  $ton \cdot km^{-2} \cdot day^{-1}$ . This reduction was mainly attributed to the stable atmospheric stratification and weaker easterly winds during winter (Fig. 5). Analysis of HYSPLIT backward trajectories showed that the dusts in "East irrigation" region primarily originated from the short-range transport in the lower atmosphere from the east and northeast (Fig. 9c, 9d). The Gurbantunggut Desert and TD served as the largest two sources, which contributed about 40% and 27% of the occurrences, respectively. Above 3 km, the amount of east-to-west dust flux presented a trend of increasing first then decreasing, which was the synergy of the strengthening westerly winds and the decreasing dust concentration. The seasonal dust flux in layers of 4-6 km could exceed 50  $ton \cdot km^{-2} \cdot day^{-1}$ . The average west-to-east dust flux above 2 km was larger in winter and spring, with the averages of 34.4 and 26.0  $ton \cdot km^{-2} \cdot day^{-1}$ , respectively. Iwasaka et al. (2008) conducted some balloon observations in Dunhuang under calm weather conditions from 2001 to 2004 and found that, in the free troposphere, the horizontal flux of Asia background dust driven by westerlies ranged from 5.3 to  $68.7 \, ton \cdot$  $km^{-2} \cdot day^{-1}$ , and it was approximately 50 ton  $\cdot km^{-2} \cdot day^{-1}$  at altitudes of 4-6 km. These in-situ measurements were in good agreement with our satellite-based estimation.

Figure 9. (a) Vertical distribution of DFR\_U; (b) the average values of DFR\_U in the lower atmosphere (below 2km, dominated by easterly winds) and the upper atmosphere (dominated by westerly winds); (c) the cluster statistics and (d) transport altitude variations from the HYSPLIT 72h backward trajectory analysis in TD (40-42.5°N, 86°E) in spring.

# 3.4 Dust backflow in TD

To quantify the amount of dust transported to TP, four cross-sections of S1-S4 (Fig. 1) were selected along the main dust transport routes toward TP. S1 and S2 accounted for the northerly dust flux to TP, which represented the TD-Kunlun Mountains route (via the northwestern slope of TP) and the TD/GD-Qaidam Basin route (via the northeastern slope of TP), respectively. S3 and S4 considered the dust flux from TH along the western entrance and the southern slope of TP, respectively. It was important to note that the total dust flux toward TP calculated here included dust not only from these three main deserts but also potentially from other sources, such as deserts in Africa and the Middle East. In addition, due to the high elevation, intense solar radiation, and blocking moisture mountains, the climate in the Qaidam Basin is extremely arid (Jiang et al., 2022), which makes it a major dust source region in TP. The dust lifted here could not only influence the local environment, but also





continually affect the downstream cities, such as Xining and Lanzhou, through westerly winds (Yang et al., 2024). Therefore, the X1 and X2 cross-sections (Fig. 1) were selected to quantify the dust contribution of the Qaidam Basin to the downstream regions.

As shown in Table 3, the dust route via the Qaidam Basin (S2) was the most important contributor, with an annual total of 7000  $ton \cdot km^{-1}$ . The main dust transports along this route were in winter and spring, with seasonal totals of 4233 and 2281  $ton \cdot km^{-1}$ , respectively. The annual dust amount to the TP via northern slope of the Kunlun Mountains (S1) was 2409  $ton \cdot km^{-1}$ .  $km^{-1}$ , which is only about 34% of the amount at S2. Dust transport along this route was primarily occurred in spring and summer, with seasonal total exceeding 1400  $ton \cdot km^{-1}$ . The sum of dust reaching the TP via the S1 and S2 paths was about  $10.2 Tg \cdot yr^{-1}$ , which was close to the  $8.0 Tg \cdot yr^{-1}$  estimated by Hu et al. (2020) from WRF-Chem simulations. Additionally, the dust imported from S4, dominated by westerly winds, contributed an annual total of 3861  $ton \cdot km^{-1}$ , which was the second largest contributor. This route also showed net incomes in all seasons. At the southern slope (S3), due to the obstruction of Himalaya, it appeared as the smallest annual dust contributor of 549  $ton \cdot km^{-1}$ . Dust transport along this route was primarily occurred in spring, with a seasonal total of 392  $ton \cdot km^{-1}$ , which was more than four times of that in summer and autumn. This was related to the extensive updrafts over the Indian plains in spring (Fig. 8) and the helpful southerly winds between  $90^{\circ}$  -100° E (Fig. 5), of which the former facilitated the transport of dust beyond the desert under the westerly winds. For the dust flux in the Qaidam Basin, it was found that the west-to-east dust transport occurred throughout the year, with the net total annual amount reaching 6063  $ton \cdot km^{-1}$  after considering both the income and outcome. The highest contribution occurred in spring, which accounted for more than half of the annual total, which was closely related to the strong surface winds in the northeastern plateau during spring (Fig. 5a). In short, our satellite-based estimation confirmed that the Qaidam Basin was a significant source of dust aerosols for the downstream regions.

**Table 3.** Seasonal and annual totals of column dust flux (unit:  $ton \cdot km^{-1}$ ) toward the TP across the four sections of S1-S4 and the contribution of Qaidam Basin to the downstream regions due to westerly winds. (The dust flux at sections of S1-S3 only considered the north-south direction, while at S4 only considered the east-west direction. X1 and X2 represented the western and eastern calculation boundaries for the Qaidam Basin, respectively. The negative values represented the exporting dust flux.)

|                   | MAM       | JJA       | SON       | DJF       | Annual     |
|-------------------|-----------|-----------|-----------|-----------|------------|
|                   | (92 days) | (92 days) | (91 days) | (90 days) | (365 days) |
| S1 (northwestern) | 1418      | 1460      | -419      | -50       | 2409       |
| S2 (northeastern) | 2281      | 526       | -40       | 4233      | 7000       |
| S3 (southern)     | 392       | 94        | 89        | -26       | 549        |






| S4 (western)             | 1894 | 349 | 434  | 1185 | 3861 |
|--------------------------|------|-----|------|------|------|
| X1, X2<br>(Qaidam Basin) | 3064 | 886 | 1158 | 955  | 6063 |

# 3.5 Diurnal variations of dust transport

The diurnal variations in the vertical distribution and column-averaged intensity of dust mass flux during dusty days at the four cross sections of S1 to S4 (in Fig. 1) were shown in Fig. 10. The positive and negative values of dust mass flux indicated the incoming and outgoing of dust as to TP, respectively. It was important to note that precipitation on the TP occurred mainly from May to October, accounting for 91% of the annual total (Wan et al., 2017). The effects of cloud cover might result in more missing data at the S3 and S4 sections during summer and autumn, which potentially introduced underestimation of the dust transport flux during these seasons. The dust mass flux across the four sections exhibited relatively consistent diurnal features. Dust transported to the TP mainly occurred during the daytime, particularly from midday to afternoon, which was generally coincided with the period of high wind speed toward the TP (Fig. 11). In the northern TP, dust transport was stronger in the afternoon compared to the morning, while in the southern and western TP, the transport was more intense in the morning. However, there remained pronounced seasonal differences. First, the strongest transport periods varied across seasons. In spring, summer, and autumn, the peak transport mainly occurred during the daytime, while in winter it predominantly happened at night. Furthermore, the dominant heights for strong dust transport varied by season. In spring, the dust activity was most intense, and the dust transport occurred throughout the entire atmospheric column. In summer, precipitation suppressed the raising and mixing of dust in vertical, leading to a slight decrease in dust transport. In autumn, enhanced atmospheric stability confined the dust transport in the lower atmosphere, while in winter stronger winds at higher altitudes increased dust transport in the upper air. The pattern of diurnal dust transport peaks varied with the seasons. During spring, summer, and autumn, dust transport generally followed a unimodal pattern with peaks occurring during the daytime, whereas in winter it exhibited a bimodal pattern (e.g., at S2 and S4), with peaks occurring in both daytime and nighttime. The nighttime secondary peak was closely associated with the strong high-altitude winds. Fig. 11(m-p) also showed that intense westerly winds carried large amounts of dust from surrounding regions, such as Central Asia and the deserts of Africa, into the TP. The strongest threehour dust flux in dusty days occurred at the western entrance of the TP, with peaks of 5068 and 3297  $kg \cdot km^{-2} \cdot hr^{-1}$  at 05-07 UTC in spring and at 23-01 UTC in winter, respectively.

Figure 10. The diurnal variations in the vertical distribution and column-averaged intensity of dust mass flux at cross-sections S1-S4 during dusty days. (The positive and negative values indicated the dust transport towards and away from TP, respectively. The dashed line denoted the 12:00 local time.)

**Figure 11.** The diurnal variations of wind speed at cross-sections S1-S4 during dusty days. (The positive and negative values indicated the wind flowing towards and away from TP, respectively. The dashed line denoted the 12:00 local time.)

# 3.6 Estimation of dust exposure




Dust transport could have significant adverse effects on human society. When crossing highly industrialized and polluted areas in China, dust could absorb heavy metals, dioxins, radioactive isotopes, and other harmful substances (Goudie, 2014; Sprigg et al., 2014). Short-term impacts included effects on the respiratory, skin, eye, cardiovascular, and neurological systems, while long-term effects might affect the health of fetuses, infants, and pregnant women, and potentially lead to infectious diseases (Aghababaeian et al., 2021; Opp et al., 2021). Studies have shown that dust weather increased hospitalizations for respiratory diseases (Tao et al., 2012), raised measles incidence (Ma et al., 2017), and contributed to a 4.92% increase in overall urban mortality (Chen et al., 2004). Furthermore, dust deposition could impact crop growth, resulting in a 28% decrease in cotton yields (Zia-Khan et al., 2015). In northwestern China, wind erosion caused annual economic losses of ~3 billion yuan (Zhao et al., 2022b). Therefore, we quantified and assessed the potential impacts of dust on human society in the study area based on satellite-based calculations, which could offer a more realistic evaluation compared to conventional model simulations. Fig. 12 showed the spatial distribution of dust exposure for human societies based on CATS observations, considering both




population density and the air temperature effect. As shown in Fig. 12, three regions within the study area warranted a significant attention (dust exposure  $>5 mg \cdot m^{-2}$ ). The first is the Indian subcontinent, extending from TH through the Indo-Gangetic Plain to the foothills of the Himalayas, which was characterized by very high dust exposure ( $> 10 \text{ mg} \cdot \text{m}^{-2}$ ). The second is the downstream areas of TP, extending from the Sichuan Basin to the large areas of the central China, where had many megacities. Although having lower dust loading compared with deserts, the urban dust exposure impact was much higher due to the denser population. The third region is the northern and southern edge of Tarim Basin, where lots of dust events appeared due to airflow convergence and uplift. The population density is high near the Tianshan Economic Belt, where the impact is significant throughout the year. From a seasonal perspective, the dust exposure was higher in spring and summer. Particularly in the Indian subcontinent, the dust exposure in summer was very high under the combined effect of high temperature and strong dust storms, with values even exceeding 40 mg  $\cdot$  m<sup>-2</sup>. In the downstream cities of the TP, the area of dust exposure  $>5 mg \cdot m^{-2}$  was at its widest in spring, followed by winter and summer, and the smallest in autumn. During winter and spring, the dust impact was relatively larger in the central and eastern regions than the other parts of the TP. Studies had shown that in areas such as the Hetian and Aksu regions of the TD (Zhao et al., 2011), India (Gupta et al., 2020), and the northern China (Shen et al., 2025), excessive dust exposure had been confirmed to pose pathogenic and even fatal risks, presenting a serious threat to human health. Therefore, implementing effective air quality monitoring and timely health warning systems in these areas is crucial to significantly reduce health risks and economic losses caused by dust pollutions. Furthermore, effective comprehensive policies for desertification control should be developed under scientific guides, including land use management, water resource management, and vegetation restoration. Overall, addressing the impact of dust on human societies remained a long-term challenge that required interdisciplinary collaboration and policy intervention.

**Figure 12.** The distributions of dust exposure.

# 4 Conclusions



This study utilized the observations from CATS and CALIPSO, combined with datasets of DustCOMM, MERRA-2, CHAP and ERA5, to depict the seasonal and diurnal variation characteristics of dust transport around TP (Tibetan Plateau). The main conclusions were as follows:

- The four datasets of CATS, MERRA2, CHAP and DustCOMM depicted a similar pattern of dust distribution in the study region. All datasets showed consistent locations of high dust concentration, which are the southern edge of TD (Taklimakan Desert), the southwestern TH (Thar Desert), the northern GD (Gobi Desert), and the Qaidam Basin in TP (Tibetan Plateau). The dust concentration calculated from CATS showed a correlation coefficient of 0.4-0.6 with the ground-based observations and reanalysis datasets.
- Two distinct dust transport belts are located respectively on both the northern and southern sides of TP. The northern branch originated from the TD and extended to the northeastern China, while the southern one began from the TH and stretched eastly to 100° E. The coverage and total amount of dust transport are at their maximum during spring for both dust belts. As shown in Fig. 12, the annual TP-ward dust fluxes via the northern slope were 2409 ton · km<sup>-1</sup> and 7000 ton · km<sup>-1</sup> through the Kunlun Mountains route and the Qaidam Basin route, respectively. The TP-ward dust fluxes from the western entrance and the southern slope were 3861 ton · km<sup>-1</sup> and 549 ton · km<sup>-1</sup>, respectively. CATS observations demonstrated that the Qaidam Basin was an important dust source for the downstream cities, with an annual exporting dust flux of 6063 ton · km<sup>-1</sup>, in which more than half was contributed in spring.
  - A dust backflow from east to west existed all year round in the lower air over the eastern TD due to the favorable terrain and special weather systems. The seasonal flux rate of the dust backflow ranged from 7.4 to 83.1  $ton \cdot km^{-2} \cdot day^{-1}$ , with the maximum in spring.
  - The diurnal variations of dust entering the TP at different altitudes were analyzed along four major transport routes. The dust fluxes were more intense from midday to afternoon, which was partly favored by the wind speed. The dust flux showed pronounced seasonal variations both in vertical distribution and transport intensity. It was strongest in spring both for the total column and different altitude layers, while it primarily concentrated into the lower atmosphere in autumn. Dust flux during winter showed a bimodal pattern, one of which appeared at night.
  - A simply evaluation of the potential impacts of dust invasion on human society showed that the Indian subcontinent was the most severely affected area, followed by the cities in the downstream areas of the TP and the northern and southern edges of the Tarim Basin.
- 455 Data availability. The CALIPSO, CATS, MERRA-2 GPWv4 available website: and data are at https://search.earthdata.nasa.gov/. The DustCOMM data can be downloaded from website: https://zenodo.org/records/2620475. The CHAP data are available at: https://www.geodata.cn. The particulate matter observation data is downloaded from http://www.cnemc.cn/. The ERA5 data are available at https://cds.climate.copernicus.eu/.

Author contributions. Xiaofeng Xu conceived the overall idea, designed the methodology, conducted formal analysis and revised the manuscript. Zixu Xiong curated the data, prepared the original draft, and created visualizations. Jianming Gong and Huilin Zhang contributed to visualization and investigation. Tianliang Zhao and Qing He commented on the manuscript.

Competing interests. The authors declare no competing interests.

Acknowledgements. The authors are grateful to the data providers. We thank NASA Earthdata for providing the CATS, CALIPSO, MERRA-2, and GPWv4 datasets, the European Centre for Medium-Range Weather Forecasts (ECMWF) for the ERA5 dataset, the Aerosol-Climate Interactions research group for the DustCOMM dataset, the National Earth System Science Data Center for the CHAP dataset, and the China National Environmental Monitoring Center for particulate matter observation data. This work was funded by the National Natural Science Foundation of China (grant number: 42030612).

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
