# Peer review of "Diurnal and Seasonal Variations of Dust Transport around the Tibetan Plateau: Insights from Multi-Source Observations"

_EGUsphere, 2025_

## Author Comment (AC1)

**Response to Referee #2' Comments**

**Response to Major comments**

1) Dust concentration and dust flux were derived through inversion from multiple datasets, which inherently involve various uncertainties that may affect the reliability of the results. These include the spatiotemporal sampling limitations of CATS, potential errors in the mass extinction efficiency (MEE) values, and uncertainties related to wavelength extrapolation between CALIPSO and CATS. It is recommended that the authors add a dedicated section (either in the main text or as an appendix) to qualitatively or quantitatively discuss these error sources and their potential impacts on the key conclusions. This would significantly improve the completeness and credibility of the study.

**Response:** Thank you for the suggestion. We have added a dedicated Section 3.7 in the main text to systematically address the retrieval uncertainties of dust concentration and flux. The discussion covered uncertainties arising from the selection of the mass extinction efficiency (MEE), CATS sampling and cloud-aerosol discrimination, extinction wavelength conversion, and wind fields. Based on this analysis, we estimated that the overall uncertainty in dust mass concentration is approximately 10–20%, while that in dust flux is about 25–45%.

2) The authors used the wavelength ratio from CALIPSO at 1064/532 nm to convert the CATS 1064 nm extinction data. The accuracy of the dust extinction coefficient derived through this approach remains unclear. It is advisable to further validate the quality of the CATS dust extinction inversion over the study region, for instance, by comparing with independent measurements or other relevant datasets.

**Response:** Thanks for the suggestion. We have included a comparison between CATS and the CALIPSO L3 product at 532 nm to evaluate the practical performance of the wavelength conversion.

As illustrated in Fig. S6 (in the supplement), the springtime vertical profiles of dust extinction coefficients over three representative regions demonstrated that CATS captured the main dust distribution patterns, their typical decrease with height, and relevant regional differences. For instance, dust over the Thar Desert was strongest near the surface and declines rapidly with height, whereas a distinct elevated dust layer around ~3 km above sea level was present over the Taklamakan Desert. However, CATS exhibited a slight positive bias in the mid-troposphere (~4–6 km) relative to CALIPSO. The full-year scatter statistics (Fig. S7 in the supplement) further showed good regional-scale agreement between the two datasets, with high correlation (r = 0.88) and low errors (Mean Bias ≈ 0.01 $km^{-1}$, MAE ≈ 0.01 $km^{-1}$, RMSE ≈ 0.02 $km^{-1}$). These results demonstrated that CATS retrievals reliably captured the spatiotemporal variability and relative magnitude of dust extinction, and were suitable for analyzing dust spatial distribution, seasonal cycles, and diurnal characteristics.

[Figure]

**Figure S6.** Comparison of springtime $\sigma_{D\_A}$ profiles at 532 nm between CATS and CALIPSO.

[Figure]

**Figure S7.** Comparison of annual-mean $\sigma_{D\_A}$ at 532 nm between CATS and CALIPSO Level 3 data. The number in parentheses following the total sample size indicated the percentage of samples retained after removing outliers using the interquartile range (IQR) method.

3)The formula used for the "Dust Exposure" metric appears relatively simplistic and lacks a solid scientific rationale. The authors should provide further justification for the theoretical basis of this indicator and clarify its applicable scope and limitations.

**Response:** Thanks for the comment. We agree that this metric was overly simplistic and lacked a solid theoretical basis. Therefore, we have removed this part from the revised manuscript.

4)While the study estimates dust input from surrounding deserts to the plateau via cross-sectional flux integration, as well as the contribution from the Qaidam Basin to downstream regions, the potential influence of locally emitted dust from within the Tibetan Plateau itself is not sufficiently addressed. Differentiating between the contributions of local soil erosion and long-range transported dust would add significant value to the analysis.

**Response:** Thanks a lot for the suggestion. We agree that it is important to distinguish between local dust emissions on the Tibetan Plateau and dust transported from distant sources. We would like to clarify that this study focused on quantifying two dust-related processes from a flux perspective. To do this, we first used boundaries S1–S4 to estimate the inflow of externally transported dust into the Tibetan Plateau. We then treated the Qaidam Basin as a major local dust source and used transects X1–X2 to estimate its net downwind export. In this way, our analysis already accounted for both external dust inflow and local dust sources within the Plateau.

Our results indicated clear annual differences in external dust inflow across the boundary transects S1–S4. The northern pathways (S1+S2) serve as the dominant route, with an inflow magnitude of about 10.2 $Tg \cdot yr^{-1}$. For local dust sources within the Plateau, the Qaidam Basin showed an annual net export of about 6063 $ton \cdot km^{-1}$, as estimated from transects X1–X2, with the largest contribution occurring in spring. Overall, these results indicated that—at the transport/export flux scale examined here—the dust budget over the Tibetan Plateau was influenced both by long-range inflow from surrounding deserts and by outflow from internal dust-source regions such as the Qaidam Basin.

We also highlight that local dust emissions from wind erosion require more precise parameterization and inventory support. Our satellite-based approach instead characterized column dust loading, advective transport, and net regional export. Therefore, within the current framework, we quantified the internal dust contribution from the Plateau in a representative manner by estimating the net export from local source regions such as the Qaidam Basin. This approach provided flux-based evidence that allowed direct comparison between local contributions and external transport, without introducing additional assumptions from emission models. We appreciate the reviewer's point and consider it an important direction for future work, which could include incorporating emission parameterizations and further resolving dust contributions from different land-surface types within the Plateau.

**Response to Minor comments**

1)The points of innovation are currently scattered throughout the manuscript. They should be clearly and concisely summarized in both the introduction and conclusion.

**Response:** Thank you for the constructive comments. We agree that the innovation of the original manuscript was scattered across the abstract, methods, and results sections, which weakened the clarity of the scientific storyline. To address this, we have consolidated the key innovations in the introduction and conclusion of the revised manuscript and presented them in a coherent, echoing manner.

2)In several figures (e.g., Figures 5 and 7), the axis labels, legend text, and color bar scales are too small, making them difficult to read even when zoomed in. It is recommended to increase the font sizes appropriately to improve readability.

**Response:** Thank you for the reminder. We have improved the previously unclear figures, including Fig. 5 (now Fig. 4) and Fig. 7 (now Fig. 6).

[Figure]

**Figure 5.** Seasonal distribution of CATS DFRC ($kg \cdot m^{-1} \cdot day^{-1}$, a-d), and DFRC_U and DFRC_V from CATS ($kg \cdot m^{-1} \cdot day^{-1}$, e-l) and MERRA-2 (m-t) datasets during the period from March 2015 to October 2017, respectively. (Positive values of DFRC_U and DFRC_V corresponded to eastward and northward transport, respectively.)

[Figure]

**Figure 7.** Seasonal distribution of DFR_W (dust flux rates at vertical direction, $g \cdot m^{-2} \cdot day^{-1}$) at four pressure levels from CATS during the period from March 2015 to October 2017. (The arrows represent the horizontal wind field. The positive values of DFR_W corresponded to upward transport.)

3)The latitude and longitude ranges of the cross-sectional lines (S1–S4, X1–X2) in Figure 1 should be clearly labeled to enhance interpretability.

**Response:** Thank you for the suggestion. We have added the latitude–longitude ranges for each transect in the caption of Fig. 1.

[Figure]

**Figure 1.** The topography of the study aera (23-45°N, 68-115°E). S1 to S4 (S1: 36.5°N, 76.75–86.75°E; S2: 38.5°N, 88.75–101.75°E; S3: 27.75°N, 88.25–100.25°E; S4: 30.25–35.25°N, 80.25°E) were the selected boundaries for the TP-ward dust flux calculation and X1 to X2 (X1: 35–38.5°N, 90°E; X2: 35–38.5°N, 98°E) were used to quantify the dust contribution of the Qaidam Basin to downstream regions.TD, TP, TH, GD are Taklimakan Desert, Tibetan Plateau, Thar Desert and Gobi Desert. Base map data sourced from the Natural Earth dataset.

---

## Author Comment (AC2)

**Response to Referee #1' Comments**

**Response to Major comments**

1. The extinction-to-mass conversion is on shaky ground. You're dividing CATS extinction by a fixed DustCOMM MEE that's climatological, largely dry, and wavelength/size-agnostic, while the actual extinction depends strongly on size distribution, mineralogy, refractive index, and RH. With regional dust heterogeneity across Tarim/Qaidam and potential 1064 532 nm inconsistencies, this shortcut can introduce order-of-magnitude bias unless you regionalize MEE and propagate its uncertainty.

**Response:** Thank you for highlighting potential issues associated with this key step of converting extinction coefficients to dust mass concentration. We agree that the description of the DustCOMM MEE in the original manuscript was overly brief and could inadvertently suggest that a single fixed MEE constant was used for the conversion, which did not adequately convey the physical basis of this procedure or its regional variability. In fact, we used the three-dimensional MEE dataset provided by DustCOMM at 550 nm, rather than a prescribed constant value. This product was a physically grounded global dust climatology based on a multi-model ensemble that was further constrained by multi-site observations (Adebiyi et al., 2020). During the dataset construction, factors such as particle size distribution, regional variability, and vertical structure were explicitly considered. Consequently, MEE varied with both location and altitude on a $0.5° × 0.625°$ horizontal grid and across 35 pressure levels (Figs. S1–S2). Adebiyi et al. (2020) showed that the DustCOMM MEE differed from observational estimates by only ~1% on average, and that it improved upon the raw model ensemble by better capturing lower MEE near source regions and higher MEE downwind as dust was transported and aged. Therefore, this conversion did not rely on a wavelength- or size-independent "universal constant"; instead, it used an observation-constrained, spatially and vertically varying 3D MEE field. Given the lack of long-term, continuous, and representative in situ MEE measurements along the Taklamakan–Thar–Tibetan Plateau transport corridor, region- and height-resolved MEE fields could not be independently constructed from observations alone. DustCOMM effectively addressed this data gap by providing a global, three-dimensional, observation-constrained dust MEE product, representing a pragmatic balance between feasibility and reliability. Moreover, several recent studies focusing on the source and downwind regions relevant to this work (including the Taklamakan Desert, the Thar Desert, and the Tibetan Plateau) have directly adopted DustCOMM MEE climatology to quantify dust loading and transport and have reported encouraging performance (Han et al., 2022a, 2022b; Tang et al., 2024).

We acknowledge the reviewer's point that the DustCOMM product did not explicitly account for humidity effects, which could influence MEE by altering particle size and refractive index. However, for the mineral dust and free-tropospheric conditions in this study, the use of an approximately dry-state MEE was justified for the following reasons.

(1) Laboratory constraints indicated limited mass growth of mineral dust under subsaturated conditions. For Asian mineral dust types closely relevant to our study region, vapor adsorption experiments by Chen et al. (2020) showed that the water mass at RH = 90% was only ~2% of the dry dust mass for Chinese loess, Qinghai dust, and Xinjiang dust; at RH = 70% it was ~1%. This suggested that the humidity-induced mass increase was modest under subsaturated RH.

(2) The hygroscopic diameter growth factor of natural mineral dust was close to unity under subsaturated RH. Using HTDMA measurements, Koehler et al. (2009) reported that natural mineral

dust from North Africa and the southwestern United States typically exhibited diameter growth factors < 1.1 for RH ≤ 90%, whereas more pronounced growth was mainly observed for artificially generated samples produced via nebulization. This implied that hygroscopic growth of naturally emitted mineral dust was generally weak under subsaturated conditions, with impacts on geometric size and optical cross-section at only the few-percent level.

(3) Our analysis was based on the "Dust" (not "Dust mixture") classification from CATS. Based on the typically dry free-tropospheric conditions in our study region and the constrained hygroscopic growth factors, we inferred that the RH-induced enhancement of MEE under our conditions was likely only a few percent. This effect was smaller than other dominant uncertainties, such as those from the CATS extinction retrievals and wind fields.

Based on these considerations, we concluded that treating mineral dust as weakly hygroscopic and using the DustCOMM three-dimensional (approximately dry) MEE to estimate dust mass fluxes in the dry, cold free troposphere was a reasonable approximation under current observational and parameter constraints. Due to the lack of sufficient in situ measurements, developing an observationally derived, RH-dependent MEE field was not yet technically mature. Introducing an RH-dependent "wet" MEE would rely on assumptions regarding dust mixing state, soluble fraction, and vertical humidity profiles, potentially introducing new uncertainties that were difficult to quantify with available data. Therefore, we retained the dry-MEE treatment in the revised manuscript and added text in Section 3.7 (Uncertainty assessment) to explicitly clarify the physical basis and spatial heterogeneity of DustCOMM MEE, and limitations of this assumption, including its conditions of applicability and potential bias range.

[Figure]

**Figure S1.** Horizontal distribution of the DustCOMM MEE product over the study region at 581.25 hPa. The 581.25 hPa level corresponded to an altitude of approximately 4 km, which represented the principal dust transport layer downwind of the TD, Qaidam Basin, and the TP. The MEE values across the entire region generally ranged between 1.6 and 2.9 $m^2 \cdot g^{-1}$.

[Figure]

**Figure S2.** Vertical distribution of the DustCOMM MEE product over the TD (35°–42°N, 75°–90°E). The MEE values showed a gradual increase with altitude.

Adebiyi, A. A., Kok, J. F., Wang, Y., Ito, A., Ridley, D. A., Nabat, P., and Zhao, C.: Dust Constraints from joint Observational Modelling-experiMental analysis (DustCOMM): comparison with measurements and model simulations, Atmos. Chem. Phys., 20, 829–863, https://doi.org/10.5194/acp-20-829-2020, 2020.

Chen, L., Peng, C., Gu, W., Fu, H., Jian, X., Zhang, H., Zhang, G., Zhu, J., Wang, X., and Tang, M.: On mineral dust aerosol hygroscopicity, Atmos. Chem. Phys., 20, 13611–13626, 10.5194/acp-20-13611-2020, 2020.

Han, Y., Wang, T., Tang, J., Wang, C., Jian, B., Huang, Z., and Huang, J.: New insights into the Asian dust cycle derived from CALIPSO lidar measurements, Remote Sens. Environ., 272, 112906, https://doi.org/10.1016/j.rse.2022.112906, 2022a.

Han, Y., Wang, T., Tan, R., Tang, J., Wang, C., He, S., Dong, Y., Huang, Z., and Bi, J.: CALIOP-Based Quantification of Central Asian Dust Transport, Remote Sensing, 14, 1416, 2022b.

Koehler, K. A., Kreidenweis, S. M., DeMott, P. J., Petters, M. D., Prenni, A. J., and Carrico, C. M.: Hygroscopicity and cloud droplet activation of mineral dust aerosol, Geophysical Research Letters, 36, https://doi.org/10.1029/2009GL037348, 2009..

Tang, J., Wang, T., Han, Y., Zhang, X., Tan, R., Dong, Y., He, S., Abdullaev, S. F., and Amonov, M. O.: Dominating Remote Source and Its Potential Contribution of Airborne Dust Over the Tibetan Plateau, Geophysical Research Letters, 51, e2024GL111178, https://doi.org/10.1029/2024GL111178, 2024.

2.    The reconstruction of a 3-hour diurnal cycle from CATS observations is statistically invalid. CATS data cover only a short time period and have uneven local-time sampling. Using a 60-day moving window does not compensate for aliasing or sampling bias. Without showing the local-time coverage distribution and weighting scheme, the derived "typical day" is more likely an artifact of orbital sampling than a real physical signal.

**Response:** Thanks for the careful consideration on the statistical validity of reconstructing the

diurnal cycle of dust from CATS observations. We agree that the three-year CATS record and its uneven local-time sampling presented clear limitations. Consequently, deriving a 3-hourly representative diurnal cycle was statistically constrained, and the use of a 60-day moving window could not entirely eliminate sampling aliasing and bias. We did not state clearly enough about these limitations in the original manuscript, and we are grateful to the reviewer for pointing it out.

We would like to clarify that the purpose of this study is not to reproduce the "true" 3-hourly diurnal evolution for any specific day. Instead, given the inherent sampling limitations of spaceborne lidar, our objective was to extract the climatological local-time dependence of dust from the full CATS record. It is currently not yet feasible to obtain complete, multi-year, all-weather diurnal sampling of dust vertical structure from active spaceborne sensors, especially for a large coverage. Since continuous diurnal sampling from space is currently unattainable, we employ seasonal/ climatological compositing to extract statistical diurnal tendencies—a methodology also used in other CATS-based diurnal studies of clouds and aerosols (e.g., Noël et al., 2018; Yu et al., 2021; Li et al., 2023; Wang et al., 2024). Following this climatological framework, we derived two CATS dust extinction products under the assumption that seasonal characteristics are regionally consistent: (i) a climatological dust extinction (total dust extinction divided by all valid aerosol retrievals), representing the mean state conditioned on aerosol detection; and (ii) a dust-event dust extinction (total dust extinction divided by valid dust retrievals), representing the mean under dust conditions. The former was used to estimate climatological dust mass concentration and transport fluxes, while the latter specifically quantified fluxes during dust events. Thus, the "typical diurnal cycle" obtained here should be viewed as an initial, observationally constrained estimate—a practical compromise under current sampling limitations. We agree that in future studies, with more complete observational coverage, it should employ rigorous sampling diagnostics and joint inversion to approach a more instantaneous representation of the diurnal cycle.

In the revised manuscript, we have added relevant citations, expanded the limitation section to cover sampling and aliasing, and emphasized that our results reflect climatological local-time tendencies from CATS, not true instantaneous cycles. Accordingly, we suggest that the derived diurnal patterns be regarded as qualitative or semi-quantitative references, not as exact depictions of real-time behavior.

Li, Y., Li, J., Xu, S., Li, J., He, J., and Huang, J.: Diurnal variation in the near-global planetary boundary layer height from satellite-based CATS lidar: Retrieval, evaluation, and influencing factors, Remote Sensing of Environment, 299, 113847, https://doi.org/10.1016/j.rse.2023.113847, 2023.

Noel, V., Chepfer, H., Chiriaco, M., and Yorks, J.: The diurnal cycle of cloud profiles over land and ocean between 51° S and 51° N, seen by the CATS spaceborne lidar from the International Space Station, Atmos. Chem. Phys., 18, 9457–9473, 10.5194/acp-18-9457-2018, 2018.

Wang, J., Pan, H., and An, D.: Seasonal vertical distributions of diurnal variation of ice cloud frequency by CATS measurements over a global region (51°S-51°N), Journal of Atmospheric and Solar-Terrestrial Physics, 258, 106222, https://doi.org/10.1016/j.jastp.2024.106222, 2024.

Yu, Y., Kalashnikova, O. V., Garay, M. J., Lee, H., Choi, M., Okin, G. S., Yorks, J. E., Campbell, J. R., and Marquis, J.: A global analysis of diurnal variability in dust and dust mixture using CATS observations, Atmos. Chem. Phys., 21, 1427–1447, 10.5194/acp-21-1427-2021, 2021.

3.    The treatment of the near-surface layer is oversimplified. Assuming a uniformly mixed layer

of 0–180 m ignores sharp vertical gradients under stable conditions or during dust storms. This can strongly distort the surface dust concentration and any comparison with ground-based PM10 data. A more realistic boundary-layer height estimates or sensitivity analysis is needed.

**Response:** Thanks for the reviewer's valuable comments on the treatment of the near-surface layer. We acknowledge that representing the 0–180 m layer as uniformly mixed was an idealized simplification, as strong vertical gradients could occur under stable stratification or during intense dust events. This methodological choice inevitably introduced uncertainty, affecting both the derived surface concentrations and their validation against ground-based $PM_{10}$ observations.

However, it should be noted that in this study the use of the 0–180 m near-surface layer was mainly motivated by the inherent limitations of spaceborne lidar retrievals. When retrieving near-surface signals, spaceborne lidars such as CATS and CALIOP frequently encountered challenges including surface return contamination, low signal-to-noise ratios, and cloud interference. These unfavorable conditions often resulted in data gaps (NaN) or unreliable (anomalously high or low) extinction coefficients in the lowest range bins. It was thus difficult to reliably represent surface concentrations using the single lowest layer alone. Some practical precedents have proven the effectiveness of such layer mean method, including Toth et al. (2019), who used the 100–1000 m mean CALIOP extinction to avoid surface reflection, and Han et al. (2022), who treated the 0–300 m mean as surface concentration. Considering these constraints and CATS's 60 m resolution, our adoption of the 0–180 m layer was a pragmatic compromise between data reliability and near-surface representativeness.

To evaluate the climate-scale validity of this near-surface approximation, we conducted a systematic comparison between CATS retrievals and reanalysis data. Fig. S3 presented the seasonal-mean relationship between MERRA-2 surface dust mass concentration (DMCS) and the CATS-derived near-surface concentrations. Upon removing a small set of extreme outliers (<7% of total samples) via the IQR method, the two datasets show strong agreement in their seasonal spatial patterns, with correlation coefficients ranging from 0.54 to 0.74 across the four seasons (mean ≈ 0.67). Compared to MERRA-2, CATS showed a modest positive bias in spring (+13 $\mu g \cdot m^{-3}$) and negative biases of 18–27 $\mu g \cdot m^{-3}$ in other seasons, yielding an annual mean bias of –10 to –20 $\mu g \cdot m^{-3}$. Seasonal MAE and RMSE were 41–68 $\mu g \cdot m^{-3}$ and 57–98 $\mu g \cdot m^{-3}$, respectively. Relative to typical dust concentrations (150–250 $\mu g \cdot m^{-3}$) in source and downwind areas, this MAE corresponded to about 20–40% of the mean, and could be larger under conditions with pronounced gradients or specific meteorology. Fig. S4 further compared the seasonal relationships between CATS 0–180 m near-surface dust concentration, MERRA-2 DMCS, and ground-based $PM_{10}$-$PM_{2.5}$ data. After IQR-based outlier removal (<7%), CATS showed moderate-to-high correlation (r ≈ 0.56–0.68) and generally comparable magnitude with $PM_{10}$-$PM_{2.5}$, but with an overall negative bias of ∼–20 $\mu g \cdot m^{-3}$ (i.e., slight overestimation in spring and underestimation in other seasons). It should be noted that $PM_{10}$-$PM_{2.5}$ was only an approximate indicator of dust concentration, as it included non-dust coarse particles and excluded fine dust (accounted for in $PM_{2.5}$). As an additional reference, we also examined the relationship between MERRA-2 dust concentration and $PM_{10}$-$PM_{2.5}$. MERRA-2 achieved higher correlations in all seasons (r ≈ 0.65–0.86) with lower MAE and RMSE than CATS, though it similarly showed a systematic negative bias of about –9 to –19 $\mu g \cdot m^{-3}$. This bias likely arose from the representativeness gap between station observations and model/satellite grids, along with the influence of near-surface vertical gradients.

Overall, treating the 0–180 m layer as uniformly mixed was a simplification that might underestimate near-surface gradient effects, particularly during stable stratification or intense dust events. Nevertheless, after removing extreme outliers, the CATS near-surface dust concentrations demonstrate substantial consistency with both MERRA-2 and ground-based PM$_{10}$-PM$_{2.5}$ in seasonal-mean spatial patterns and magnitude. The uncertainty was comparable to that found in similar CALIOP/CATS retrieval studies. Hence, we consider the 0–180 m layer to be a quantitatively viable near-surface representation for our climatological and regional-mean analysis, with uncertainties that were both acceptable and controllable. In the revised manuscript, we have explicitly stated the limitations of this approximation. We further note in the outlook that future improvements in near-surface dust retrieval will require both more accurate boundary-layer height estimates and systematic sensitivity analyses regarding the choice of layer thickness.

Han, Y., Wang, T., Tang, J., Wang, C., Jian, B., Huang, Z., and Huang, J.: New insights into the Asian dust cycle derived from CALIPSO lidar measurements, Remote Sensing of Environment, 272, 10.1016/j.rse.2022.112906, 2022.

Toth, T. D., Zhang, J., Reid, J. S., and Vaughan, M. A.: A bulk-mass-modeling-based method for retrieving particulate matter pollution using CALIOP observations, Atmos. Meas. Tech., 12, 1739–1754, 10.5194/amt-12-1739-2019, 2019.

[Figure]

**Figure S3.** Comparison of CATS 0–180 m DMCS and MERRA-2 DMCS at seasonal-mean scales. The number in parentheses following the total sample size indicated the percentage of samples retained after removing outliers using the interquartile range (IQR) method.

[Figure]

**Figure S4.** Comparison of surface $PM_{10}$-$PM_{2.5}$ observations with (a–d) CATS 0–180 m DMCS and (e–h) MERRA-2 DMCS at seasonal-mean scales. The number in parentheses after the total sample size indicated the percentage of samples remaining after outlier removal using the interquartile range (IQR) method. PM10-PM2.5 means the difference of PM10 and PM2.5.

4.    The flux integration across regional boundaries is not well justified. The authors multiply a sparsely sampled DMC profile by ERA5 winds and integrate over large sections, implicitly assuming representativeness and mass conservation that are never verified. There are no flux divergence checks or control-volume closure tests, and the temporal normalization (instantaneous, seasonal, or annual) is unclear. The resulting flux magnitudes, comparable to or exceeding full chemistry models, are likely overestimated.

**Response:** We are grateful to the reviewer for the insightful suggestions. We recognize that the original manuscript did not adequately describe the boundary flux integration method, potentially leading to unclear expression. In this work, we did not rely on extrapolating a single, spatially or temporally sparse dust mass concentration (DMC) profile, but the DMC profiles were obtained from gridded, averaged dust extinction coefficient data at $0.5° \times 0.5°$ horizontal and 60 m vertical resolution. Then we composited DMC seasonally and diurnally along each boundary to obtain representative mean profiles and combined them with coincident ERA5 winds. The flux was then integrated vertically and along the boundary direction. As a result, the fluxes reported are seasonal means, and then converted to annual totals ($Tg \cdot yr^{-1}$) based on seasonal length, and do not represent instantaneous transports. To evaluate the reliability of these flux estimates, we compared them quantitatively with existing observational and model-based studies. Our estimated seasonal-mean westerly-driven dust flux over northwestern China (2.5–7 km layer) fell between 6.8 and 34.4 $ton \cdot km^{-2} \cdot day^{-1}$. This range is consistent with the 5.3–68.7 $ton \cdot km^{-2} \cdot day^{-1}$ reported by Iwasaka et al. (2008) for free-tropospheric Asian dust based on Dunhuang balloon soundings (2001–2004), which also noted a flux of ~50 $ton \cdot km^{-2} \cdot day^{-1}$ at 4–6 km. Furthermore, our estimated total dust transport to the Tibetan Plateau via the S1 and S2 transects is ~10.2 $Tg \cdot yr^{-1}$, consistent with the 8.0 $Tg \cdot yr^{-1}$ simulated by Hu et al. (2020) using WRF-Chem. Though marginally higher, our estimates were comparable in magnitude, with bias within the overall uncertainty of the retrieval and sampling approach. Our method likely provided an upper-bound flux estimate for two reasons: (1) satellite retrievals favored strong dust events, leading to a seasonal-mean DMC biased high; and (2) uncertainties in the DMC retrieval parameters and potential systematic positive bias in ERA5 wind speeds within the transport layer could further raise the estimated flux.

     We thank the reviewer for proposing a control-volume analysis to examine flux divergence and mass conservation. While such a closed budget would indeed be valuable under ideal conditions, it relied on spatially and temporally continuous 4D dust and wind data across the entire region—a requirement which may exceed the current capabilities of satellite-based sampling. Accordingly, in the revised manuscript we emphasize that the boundary fluxes are intended as a diagnostic indicator of key transport routes and their seasonal changes, not as a rigorously closed mass budget. We have explicitly listed this distinction as a methodological limitation.

Hu, Z., Huang, J., Zhao, C., Jin, Q., Ma, Y., and Yang, B.: Modeling dust sources, transport, and radiative effects at different altitudes over the Tibetan Plateau, Atmospheric Chemistry and Physics, 20, 1507–1529, https://doi.org/10.5194/acp-20-1507 2020, 2020.

Iwasaka, Y., Li, J. M., Shi, G.-Y., Kim, Y. S., Matsuki, A., Trochkine, D., Yamada, M., Zhang, D., Shen, Z., and Hong, C. S.: Mass Transport of Background Asian Dust Revealed by Balloon-Borne Measurement: Dust Particles Transported during Calm Periods by Westerly from Taklamakan Desert, Advanced Environmental Monitoring, 121–135, https://doi.org/10.1007/978-1-4020-6364-0_9, 2008.

5. Uncertainty assessment is incomplete. Errors from aerosol typing, wavelength conversion, MEE selection, humidity effects, and wind fields should be propagated through each processing step. Simply reporting correlations of 0.4–0.6 with reanalysis products does not demonstrate accuracy. Confidence intervals or uncertainty ranges for all major quantities are necessary for the results to be credible.

**Response:** Thanks for the valuable comments regarding the uncertainty assessment. We fully agree that reporting correlation coefficients of 0.4–0.6 with a reanalysis product was insufficient to robustly validate our results and cannot replace a systematic uncertainty analysis. As highlighted, uncertainties arose in each step of the process, including aerosol classification, wavelength conversion, MEE selection, humidity influence, and wind-field errors.

While these errors ideally should be propagated stepwise in a formal analysis, two practical limitations prevented such a rigorous approach in this study. First, several key error sources could not be adequately represented by gridded statistical models. Aerosol typing errors, for instance, were discrete and highly scenario-specific—varying with situations like cloud-aerosol mixtures, low-level stratocumulus, or elevated dust plumes. To mitigate such issues, the CATS Cloud–Aerosol Discrimination (CAD) algorithm implemented several checks, including horizontal continuity and cloud-fraction tests. Second, the sparse and irregular sampling of satellite profiles introduced representativeness errors that are difficult to quantify with a full covariance structure for each grid cell, unlike in numerical models. Imposing a formal stepwise error-propagation framework under these limitations could yield confidence intervals with limited physical interpretability. Therefore, in the revised manuscript we have implemented an integrated uncertainty assessment for DMC, rather than attempting to propagate errors formally through each separate step (e.g., aerosol classification, wavelength conversion, MEE choice, and humidity adjustments). Specifically, we began with the final retrieved products—DMCS and dust mass column concentration (DMCC)— and performed systematic comparisons with reanalysis datasets and ground-based observations to constrain their overall uncertainty.

As shown in Fig. S3, the seasonal spatial correction coefficient between CATS and MERRA-2 surface dust concentration (DMCS) ranged from 0.54 to 0.74, with the MAE of 41-68 $\mu g \cdot m^{-3}$. Based on the typical regional dust concentration of 150-250 $\mu g \cdot m^{-3}$, this error corresponded to a relative error of approximately 20–40%. As shown in Fig. S4, the correlation between the near-surface dust concentrations retrieved by CATS (MERRA-2) and the PM$_{10}$-PM$_{2.5}$ concentrations from ground observation stations was in the range of 0.56–0.68 (0.65–0.86), with an overall negative bias of ~ –20 $\mu g \cdot m^{-3}$ (–9 ~ –19 $\mu g \cdot m^{-3}$). A further comparison of seasonal DMCC (Fig. S5) shows better correlations of 0.57–0.84 between CATS and MERRA-2, with lower bias, MAE, and RMSE than for near-surface concentrations, indicating better column-level consistency. Combined retrieval errors from both column and near-surface estimates, it suggests an overall uncertainty of ~20–40% for the multiyear-mean dust concentration retrieval. The ERA5 10 m wind speed product has been shown to agree well with surface observations with a global bias of −4.5% (Fan et al., 2021) and to outperform MERRA-2 in capturing East Asian surface wind patterns under a bias of ~0.6 m s⁻¹ (Li et al., 2025). Differences between ERA5 and other reanalyses (e.g., MERRA-2, CFSv) across altitude levels are also generally small (Wu et al., 2024), supporting its reliability for multi-level analysis. Given that the regional mean systematic bias is < 5% and the random deviation at different vertical levels is < 0.5 $m \cdot s^{-1}$, we conservatively assume a random uncertainty of 10–

20% for ERA5 wind speeds in the subsequent error-propagation analysis. On this basis, the dust transport flux uncertainty was expressed as the combined effect of the mass and wind-speed terms, yielding an overall uncertainty range of approximately 25–45%. The above analysis has been incorporated into the revised manuscript.

Fan, W., Liu, Y., Chappell, A., Dong, L., Xu, R., Ekström, M., Fu, T.-M., and Zeng, Z.: Evaluation of Global Reanalysis Land Surface Wind Speed Trends to Support Wind Energy Development Using In Situ Observations, Journal of Applied Meteorology and Climatology, 60, 33–50, https://doi.org/10.1175/JAMC-D-20-0037.1, 2021.

Li, S., Wang, K., Miao, H., Zhu, X., Liu, Y., Li, J., Wang, W., Zheng, X., Feng, J., and Wang, X.: Evaluation of surface wind speed over East Asia and the adjacent ocean in three reanalyses using satellite and in-situ observations, Atmospheric and Oceanic Science Letters, 18, 100587, https://doi.org/10.1016/j.aosl.2024.100587, 2025.

Wu, L., Su, H., Zeng, X., Posselt, D. J., Wong, S., Chen, S., and Stoffelen, A.: Uncertainty of Atmospheric Winds in Three Widely Used Global Reanalysis Datasets, Journal of Applied Meteorology and Climatology, 63, 165–180, https://doi.org/10.1175/JAMC-D-22-0198.1, 2024.

[Figure]

**Figure S5.** Comparison of DMCC between CATS and MERRA-2 at seasonal-mean scales. The number in parentheses following the total sample size indicated the percentage of samples retained after removing outliers using the interquartile range (IQR) method.

6. Aerosol classification and cloud filtering are not handled carefully enough. Over complex terrain, small misclassification rates between dust and cloud layers can generate large mass errors after the extinction-to-mass step. Independent validation using AERONET or ground-based lidar

data should be provided to quantify such effects.

**Response:** Thanks for the suggestions about aerosol classification and cloud screening issues. The CATS Level-2 products were generated using a Cloud–Aerosol Discrimination (CAD) algorithm accompanied by various quality-control tests, which were designed to differentiate clouds from aerosols and to flag cases of potential cloud contamination (CATS L2O Profile Products Quality Statements, Version 3.00). In the processing, we imposed further strict screening criteria on top of the standard CATS quality controls: (1) the 1064 nm extinction QC flag must be 0 to ensure non-opaque layers and stable lidar ratio; (2) both feature type and aerosol type must equal 3 to select dust aerosols only; (3) the feature type score is confined between $-10$ and $-2$ to exclude low-confidence layers with minimal effect on aerosol optical thickness (AOD); (4) the 1064 nm extinction uncertainty must be $<10$ $km^{-1}$ for stable retrieval; and (5) the 1064 nm extinction coefficient is limited to $-0.05$–$1.25$ $km^{-1}$ to reduce averaging-induced high bias (Xiong et al., 2023). It is worth noting that CATS has incorporated multiple algorithmic improvements for cloud–aerosol discrimination and aerosol typing, including a cloud-fraction threshold, a layer-integrated vertical backscatter test, and a relative-humidity test (CATS L2O Profile Products Quality Statements, Version 3.00). These refinements helped to better differentiate optically thick clouds from intense dust/smoke plumes and reduced errors such as misidentifying elevated dust as ice cloud or thick smoke as cloud. As a result, the accuracy of aerosol–cloud separation has been substantially improved under complex cloud-field conditions.

We appreciate the reviewer's suggestion to perform independent validation using AERONET or ground-based lidar data in our study region. However, the sparse coverage of AERONET and ground-based lidar sites over the Tibetan Plateau and surrounding areas, combined with the limited number of spatiotemporal matches with CATS overpasses, made it difficult to perform a large-sample collocated statistical evaluation in our study region. Therefore, to account for these limitations and to evaluate the potential impact of classification errors on mass retrieval, we conducted an integrated assessment by synthesizing findings from multiple independent validation studies together with the indirect validation evidence presented in this paper (Figs. S3–S7). Proestakis et al. (2019) systematically compared CATS 1064 nm backscatter profiles with EARLINET ground-based lidar data in Europe and Central Asia. The study reported general agreement overall , with CATS showing negative biases of $-22.3\%$ (daytime) and $-6.1\%$ (nighttime). Yu et al. (2020) conducted a global analysis of the dust diurnal cycle using CATS and found that CATS-retrieved dust AOD agreed reasonably well with AERONET and MISR AOD in spatial distribution and seasonal variation, while also observing a tendency for CATS to underestimate near-surface dust extinction, particularly in daytime. Nowottnick et al. (2022) summarized the CATS aerosol typing algorithm and noted that globally, within the 0–4 km altitude range, the aerosol typing results from CATS and MERRA-2 were generally consistent. These independent validation studies consistently demonstrated that, CATS cloud–aerosol discrimination and dust identification carried certain systematic biases. The overall bias was typically on the order of ~10–30%, with a frequent tendency toward underestimation of dust loading. In addition, the comparisons between CATS dust concentrations, MERRA-2, and ground-based $PM_{10}$-$PM_{2.5}$ in this study—as detailed in our response to Comment 5 and Figs. S3–S5—showed moderate-to-high correlations and a consistent slight low bias. This result indirectly indicated that cloud-screening and aerosol-typing errors did not cause a significant net positive bias in our dust estimates.

Nowottnick, E. P., Christian, K. E., Yorks, J. E., McGill, M. J., Midzak, N., Selmer, P. A., Lu, Z., Wang, J., and Salinas, S. V.: Aerosol Detection from the Cloud–Aerosol Transport System on the International Space Station: Algorithm Overview and Implications for Diurnal Sampling, Atmosphere, 13, 1439, 2022.

Proestakis, E., Amiridis, V., Marinou, E., Binietoglou, I., Ansmann, A., Wandinger, U., Hofer, J., Yorks, J., Nowottnick, E., Makhmudov, A., Papayannis, A., Pietruczuk, A., Gialitaki, A., Apituley, A., Szkop, A., Muñoz Porcar, C., Bortoli, D., Dionisi, D., Althausen, D., Mamali, D., Balis, D., Nicolae, D., Tetoni, E., Liberti, G. L., Baars, H., Mattis, I., Stachlewska, I. S., Voudouri, K. A., Mona, L., Mylonaki, M., Perrone, M. R., Costa, M. J., Sicard, M., Papagiannopoulos, N., Siomos, N., Burlizzi, P., Pauly, R., Engelmann, R., Abdullaev, S., and Pappalardo, G.: EARLINET evaluation of the CATS Level 2 aerosol backscatter coefficient product, Atmos. Chem. Phys., 19, 11743–11764, 10.5194/acp-19-11743-2019, 2019. Xiong, Z., Xu, X., Yang, Y., and Luo, T.: Diurnal vertical distribution and transport of dust aerosol over and around Tibetan Plateau from lidar on International Space Station, Atmos Res, 294, 106939, https://doi.org/10.1016/j.atmosres.2023.106939, 2023.

Yu, Y., Kalashnikova, O. V., Garay, M. J., Lee, H., Choi, M., Okin, G. S., Yorks, J. E., Campbell, J. R., and Marquis, J.: A global analysis of diurnal variability in dust and dust mixture using CATS observations, Atmos. Chem. Phys., 21, 1427–1447, 10.5194/acp-21-1427-2021, 2021.

[Figure]

**Figure S6.** Comparison of springtime $\sigma_{D\_A}$ profiles at 532 nm between CATS and CALIPSO.

[Figure]

**Figure S7.** Comparison of annual-mean $\sigma_{D\_A}$ at 532 nm between CATS and CALIPSO Level 3 data. The number in parentheses following the total sample size indicated the percentage of samples retained after removing outliers using the interquartile range (IQR) method.

7. The wavelength mismatch between CATS extinction (1064 nm) and DustCOMM MEE (532 or 550 nm) is not properly treated. The study seems to apply a simple or fixed conversion without discussing the spectral dependence of MEE or its influence on retrieved mass (Kok et al., 2021; Vaughan et al., 2019; Song et al., 2022). This needs explicit scaling and sensitivity testing to ensure physical consistency.

**Response:** Thank you for pointing out the importance of wavelength matching. The most critical wavelength-related issue in this study arose from matching the CATS 1064 nm extinction retrievals with the conversion ratio typically defined for the visible band (532/550 nm). To address this, we constructed a season- and height-stratified ratio set, $ratio = median\ of\ \sigma_{D\_532}/\sigma_{D\_1064}$, using CALIOP dual-wavelength (1064/532 nm) dust extinction profiles (Fig. 2), and explicitly converted the CATS 1064 nm dust extinction to 532 nm based on this ratio set. The conversion ratio is about 1.2–1.5 in the near-surface layer and approaches ~1 at higher altitudes, reflecting altitude- and season-dependent variations in the particle size distribution and spectral dependence.

For the subsequent minor wavelength mismatch between 532 nm (dust extinction coefficient) and the 550 nm MEE from DustCOMM, we conducted the following sensitivity test.

Using the Ångström power-law approximation, we scaled the 532 nm extinction to 550 nm:

$$\alpha_{550} = \alpha_{532}\left(\frac{550}{532}\right)^{-\alpha}$$

Given that mineral dust typically exhibited weak to moderate spectral dependence in the visible wavelength range, we adopted α = 0.5–1.0 as the typical range and took α = 1.5 as a conservative upper bound. According to the formula above, when α = 0.5–1.0, the corresponding change in

retrieved mass was approximately −1.7% to −3.3%. Under the conservative upper bound of α = 1.5, the change reached about −5%. Hence, the uncertainty in mass estimation introduced by the wavelength difference between 532 nm and 550 nm was less than 5%, which fell within the overall retrieval uncertainty range of 20–40% for the climatological-mean dust concentration field derived in this study.

In addition, we have included a comparison between the CATS and CALIPSO L3 dust extinction coefficients at 532 nm. As shown in Fig. S6, the vertical profiles of dust extinction from the two datasets over three representative regions in spring indicated that CATS was able to reproduce the main distribution patterns and the typical height-decreasing structure of dust in the atmosphere. It also captured regional differences, such as the strongest near-surface dust loading and its rapid decreased with height over the Thar Desert, as well as the elevated dust layer around 3 km altitude over the Taklamakan Desert. Compared with CALIPSO, CATS might show a slight overestimation in the mid-troposphere (approximately 4–6 km). The full-year scatter statistics (Fig. S7) further demonstrated good consistency between CATS and CALIPSO at the regional scale, with high correlation (r = 0.88) and small biases (Mean Bias ≈ 0.01 $km^{-1}$, MAE ≈ 0.01 $km^{-1}$, RMSE ≈ 0.02 $km^{-1}$) across the annual dataset. These results indicated that CATS retrievals reliably captured the spatiotemporal variability and relative magnitude of dust extinction, and were therefore suitable for investigating the distribution, and variations of dust aerosol.

8. Deposition processes are oversimplified. Estimating basin-to-basin mass budgets using only dry deposition neglects the role of wet removal, which is significant near the Tibetan Plateau margins (Liu et al., 2023; Qian et al., 2011). The "net export" numbers could therefore be biased. Inclusion of precipitation or wet-deposition parameterizations would improve credibility.

**Response:** Thank you for highlighting this important aspect regarding the treatment of deposition processes. We agree that considering dry deposition alone might impact the estimation of net export. In this study, we adopted the framework proposed by Han et al. (2022) specifically for estimating the contribution of dust from the Qaidam Basin to downwind cities, and only included the dry deposition process. This decision was primarily based on the region's very arid climate, where the long-term mean annual precipitation was approximately 88 mm (Wang et al., 2014), making wet deposition relatively weak and the dry deposition to be the major contribution in most seasons (Fig. S8). We also fully acknowledge that in areas with relatively higher precipitation, such as the southern and western edges of the Tibetan Plateau, wet deposition played a non-negligible role in the dust mass balance (Liu et al., 2023; Qian et al., 2011). Therefore, in the revision we characterize the Qaidam Basin export flux as an upper-bound estimate based solely on dry deposition, emphasizing that wet deposition would lower the actual value. We also no longer attempt to provide a comprehensive inter-regional mass budget or infer "net export" for other areas. Instead, we focus our interpretation more cautiously on the "spatial and seasonal patterns of transport column flux". We further highlight in the outlook that the lack of constraints on wet deposition remains a key priority for future work.

Han, Y., Wang, T., Tang, J., Wang, C., Jian, B., Huang, Z., and Huang, J.: New insights into the Asian dust cycle derived from CALIPSO lidar measurements, Remote Sensing of Environment, 272, 10.1016/j.rse.2022.112906, 2022.

Liu, W., Zhao, C., Xu, M., Feng, J., Du, Q., Gu, J., Leung, L. R., and Lau, W. K. M.: Southern Himalayas rainfall as

*a key driver of interannual variation of pre-monsoon aerosols over the Tibetan Plateau, npj Climate and Atmospheric Science, 6, 57, 10.1038/s41612-023-00392-5, 2023.*

*Qian, Y., Flanner, M. G., Leung, L. R., and Wang, W.: Sensitivity studies on the impacts of Tibetan Plateau snowpack pollution on the Asian hydrological cycle and monsoon climate, Atmos. Chem. Phys., 11, 1929–1948, 10.5194/acp-11-1929-2011, 2011.*

*Wang, X., Yang, M., Liang, X., Pang, G., Wan, G., Chen, X., and Luo, X.: The dramatic climate warming in the Qaidam Basin, northeastern Tibetan Plateau, during 1961–2010, International Journal of Climatology, 34, 1524–1537, https://doi.org/10.1002/joc.3781, 2014.*

[Figure]

Figure S8. The seasonal distributions of dust dry deposition and its percentage in total deposition (dry and wet deposition) from MERRA-2 dataset of 2015-2017.

9.  Multiplying column dust mass by population and a temperature factor has no epidemiological validation or defined weighting. Without calibration or comparison to health outcome data, this index serves more as a visualization than a scientific metric.

**Response:** Thanks a lot for raising this point. We fully understand and agree that multiplying

column-integrated dust mass by population and a temperature factor lacked epidemiological validation and a clear weighting rationale. In the absence of calibration against health outcome data, such a composite index might introduce misinterpretation and did not qualify as a robust scientific metric. Accordingly, we have removed this content from the revised manuscript.

10. Similar analyses of dust pathways and seasonal transport over the Tibetan Plateau have already been reported using better constrained observational and modeling frameworks. Given the limitations of the CATS dataset, the "first 3-hour diurnal characterization" is not convincing.

**Response:** Thank you for the comment. We agree that our original claim of being the "first" was inadequately supported. We have therefore removed "first" and revised the statement to: "This study provides an extended analysis of the 3-hour diurnal variation characteristics of dust based on CATS observations."

11. Presentation issues make the analysis harder to evaluate. Units and averaging periods are sometimes missing, and the geographic boundaries of the defined sections are not clearly specified. Several figures appear over-smoothed relative to the data resolution. Adding coordinate definitions, uncertainty shading, and data-density information would help.

**Response:** Thanks for the specific suggestions concerning figure/table presentation and readability. Accordingly, we have enhanced the revised manuscript by ensuring that all units are included and that the time-averaging or compositing periods for every figure and table are explicitly stated in the corresponding captions and methodology. We have also supplemented the geographic boundaries and coordinate ranges for the S1–S4 and X1–X2 transects. To facilitate a robust assessment of our validation results, we have further provided data-density plots for the comparisons of CATS with MERRA-2, ground-based observations, and CALIPSO (Figs. S3–S7), along with uncertainty shading in extinction-coefficient plots to represent the uncertainty range. We also clarify that the smoothed profile in Fig. 9a (now Fig. 8a) was obtained via a five-point smoothing approach, and have enhanced several previously ambiguous figures.

12. Quantitative error metrics such as mean bias, normalized mean bias, or RMSE should be reported, and comparisons should be stratified by season and boundary-layer height. Including at least one well-documented dust event as a case study would make the evaluation more persuasive.

**Response:** Thanks for the suggestion. We have added a comparison of vertical profiles of dust extinction coefficients from CATS and CALIPSO at 532 nm, and have re-performed the comparisons between CATS and MERRA-2 as well as ground-based monitoring stations, with quantitative error metrics provided in Figs. S3–S7. Moreover, a case study of a major spring dust event in 2015 was added in Section 3.6. Based on CATS retrievals, we examined the dust vertical structure during this event and compared it with surface PM data. The analysis indicated that CATS retrievals were able to reasonably represent the vertical distribution and intensity of dust, and that their temporal evolution aligned with ground-based observations under changing boundary-layer and transport backgrounds.

**Response to Minor comments**

1.  The description of MEE (Eq. 2–3) lacks proper wavelength dependence and unit consistency.

**Response:** Thank you for the reminder. We have now supplemented the description of Eq. (1) by specifying the units and wavelengths of both the mass extinction efficiency (MEE) at 550 nm ($m^2 \cdot g^{-1}$) and the extinction coefficient at 532 nm ($km^{-1}$). We further clarified that the error resulting from this slight wavelength difference amounts to only a few percent and was therefore negligible (a detailed analysis was provided in our response to Major Comment 7).

2.  Figure captions are often incomplete and do not specify averaging periods or units.

**Response:** Thank you for the suggestion. We have added the averaging period and units to the figure captions.

3.  The map projections and domain boundaries (S1–S4, X1–X2) are not precisely defined, where is TD, GD, and TH, I cannot find the location.

**Response:** Thank you for the reminder. We have replaced the map projection in Fig. 1 and added the locations of TD, GD, TH, and TP to facilitate identification of the different regions. We also specified in the caption the latitude–longitude ranges of the study-area boundary and the domain boundaries (S1–S4 and X1–X2).

[Figure]

**Figure 1.** The topography of the study aera (23-45°N, 68-115°E). S1 to S4 (S1: 36.5°N, 76.75–86.75°E; S2: 38.5°N, 88.75–101.75°E; S3: 27.75°N, 88.25–100.25°E; S4: 30.25–35.25°N, 80.25°E) were the selected boundaries for the TP-ward dust flux calculation and X1 to X2 (X1: 35–38.5°N, 90°E; X2: 35–38.5°N, 98°E) were used to quantify the dust contribution of the Qaidam Basin to downstream regions.TD, TP, TH, GD are Taklimakan Desert, Tibetan Plateau, Thar Desert and Gobi Desert. Base map data sourced from the Natural Earth dataset.

4.  References to validation datasets (DustCOMM, CHAP, CNEMC) should include temporal overlap periods.

**Response:** Thank you for highlighting this important detail. We have added clarifications in the manuscript concerning the temporal overlap between the validation datasets and CATS. The periods used for CHAP and CNEMC were aligned with the CATS observation period, i.e., from March 2015 to October 2017 (this information has been added in Section 2.3). The observation period of CATS

itself has also been explicitly stated in the Data section (March 2015–October 2017). The DustCOMM dataset was generated based on the global model ensemble results during 2004–2008, among which the data periods of the WRF-Chem and IMPACT models were 2007–2016 and 2004, respectively. The dataset was further produced by integrating observational data and experimental constraints. For DustCOMM, it should be clarified that the dataset offers climatological/seasonal-mean values and did not contain monthly or daily time series corresponding to 2015–2017; thus, no strict temporal overlap interval existed. In this work, we utilized DustCOMM as a climatological reference to evaluate spatial distributions and seasonal patterns, not for time-resolved (hourly/daily) comparisons. This point has been added in the revision.

5.    The discussion mixes physical interpretation with socio-political implications, which dilutes the scientific focus.
**Response:** Thanks for the suggestion. We agree that mixing physical interpretations with socio-political implications could dilute the scientific focus of the discussion. Therefore, we have removed this part from the revised manuscript.

6.    Some figures (e.g., diurnal cycle composites) are overly smoothed and may give a misleading impression of temporal resolution.
**Response:** Thank you for pointing out this potential issue. We agree that if the diurnal composites appeared excessively smoothed, they might lead readers to perceive a temporal resolution beyond the informational limits of the data. It should be clarified that Fig. 10 (now Fig. 9) displayed a diurnal composite derived from CATS retrievals, which had a spatial grid of 0.5° × 0.5° and a vertical resolution of approximately 60 m. To increase the sample size and minimize random noise, we binned the data by UTC time and performed 3-hour averaging. For consistency, Fig. 11 (now Fig. 10), which used ERA5 fields (0.5° × 0.5°, 20 pressure levels from 1000 to 300 hPa), adopted the same 3-hour averaging to match the time bins in Fig. 10. Beyond the 3-hour bin-averaging, no further temporal smoothing or filtering was applied. The smooth appearance arose mainly from the compositing process itself, which dampened random fluctuations, and from the use of continuous color scales in the plots. To prevent any potential misunderstanding, we have revised the figure captions to clearly indicate that the figures present diurnal composites at 3-hour resolution, not raw hourly time series. This ensured that the displayed temporal resolution aligned with the data processing and could be properly interpreted by readers.

● **The new added sections (section 3.6 and 3.7)**

**3.6 Application in a dust event**

Fig. 11-12 presented a case study of an intense dust event on 31 March 2015 (UTC), aiming to evaluate the consistency between the CATS-retrieved dust properties and ground-based environmental monitoring observations. This episode was also reported by Li et al. (2018). At 22 UTC on 31 March (06 Beijing Time, BJT, on 1April), CATS overpassed the study region (Fig. 11a). At that time, the aerosol optical depth (AOD) within TD exceeded 1. The vertical profiles indicated that aerosols over both TD and TP were dominated by dust, and the DMC below 4 km in the TD exceeded 300 $\mu g \cdot m^{-3}$ (Fig. 11e, f). During this overpass, the CATS ground track approached the Aksu site located at the northern margin of the TD and the Nagqu site on the TP, providing an

opportunity for satellite–surface comparison.

During the CATS overpass, the $PM_{10}$ concentration at Aksu was only 154 $\mu g \cdot m^{-3}$ (Fig. 12a), and the nearby AOD was also clearly below 1, indicating that the near-surface air had not yet been substantially affected by the main dust plume. The near-surface dust mass concentrations retrieved from the four pixels of the trajectory closest to Aksu (approximately 115 km) ranged from 100–250 $\mu g \cdot m^{-3}$, showing a good correspondence with surface observed PM concentrations. Concurrently, weak northerly winds prevailed near Aksu (Fig. 11a). After sunrise, near-surface winds gradually shifted from northerly to southerly. By 03 UTC on 1 April (11 BJT), both the 10 m and 850 hPa wind fields near Aksu exhibited coherent southerly flow (~5 $m \cdot s^{-1}$), which favored northward transport of dust from the TD toward Aksu. Meanwhile, the boundary layer height (BLH) rose from ~250 m to ~1500 m, enhancing the mixing of the aloft dust into the near-surface layer. Under the combined effects of advection and mixing, surface $PM_{10}$ at Aksu increased rapidly to 965 $\mu g \cdot m^{-3}$ at 12 BJT (Fig. 12a). This $PM_{10}$ magnitude is comparable to the DMC ($\geq$1000 $\mu g \cdot m^{-3}$) retrieved by CATS over TD during the 06 BJT overpass (Fig. 11f), indicating that the dust transported by the southly wind from the interior of the TD desert has affected the particulate matter concentration in Aksu..These remaining deviations are mainly due to spatial collocation mismatches, temporal offsets, and the inequality of $PM_{10}$ and DMC intrinsically.

In contrast, Nagqu exhibited better spatial collocation with the CATS ground track (only ~23 km away from the closest pixel point). At 06 BJT, the observed $PM_{10}$ and $PM_{2.5}$ concentrations at Nagqu were approximately 60 and 20 $\mu g \cdot m^{-3}$, respectively, whereas the CATS-retrieved near surface DMC was 25 - 40 $\mu g \cdot m^{-3}$, showing a close agreement in magnitude (Fig. 12b, d). Overall, this case study suggested that the CATS retrievals could reasonably capture the vertical distribution and magnitude of dust, and give a comparable result of dust variation relative to the ground-based measurements.

[Figure]

**Figure 11.** Integrated analysis of a springtime dust event in 2015 based on CATS observations. (a) CATS overpass track and 10 m wind field at 22 UTC on 31 March (06 BJT on 1 April). Track

colours indicated different aerosol optical depths (AODs). Red markers denote the locations of the ground-based environmental monitoring stations of Aksu (41.18°N, 80.29°E) and Nagqu (31.48°N, 92.06°E), respectively. (b, c) Wind fields at 10 m and 850 hPa at 03 UTC on 1 April (11 BJT on 1 April). (d) Hourly boundary-layer height (BLH) at Aksu. (e, f) Vertical profiles of total extinction coefficient at 1064 nm ($\sigma_{1064}$) and DMC retrieved from CATS. Wind and BLH data are from the MERRA-2 reanalysis. The basemap is from the Natural Earth dataset.

[Figure]

**Figure 12.** (a, b) Temporal variations of $PM_{10}$ and $PM_{2.5}$ concentrations measured at Aksu (41.18°N, 80.29°E) and Nagqu (31.48°N, 92.06°E) ground-based environmental monitoring stations. (c, d) Vertical profiles of DMC retrieved from CATS along the ground track segments closest to Aksu and Nagqu during the 22 UTC overpass on 31 March 2015 (06 BJT on 1 April).

**3.7 Uncertainty and limitation**

In this study, DMC is derived from CATS extinction coefficients and subsequently used to estimate the transport fluxes. The overall uncertainty mainly arose from the choice of MEE, CATS sampling characteristics and potential cloud aerosol misclassification, wavelength conversion of extinction, uncertainties in the driving wind fields, and simplified treatment of deposition processes. Each error source was discussed below in detail.

**3.7.1 MEE**

To the DMC retrieval, we adopt the three-dimensional climatological MEE at 550 nm provided by DustCOMM. DustCOMM was a global dust climatology constructed from a multi-model ensemble that was constrained by multi-site observations, and it had a clear physical basis (Adebiyi et al., 2020). Its production explicitly accounted for particle size distributions, regional variability, and vertical structure, yielding a climatology seasonal mean MEE field that varied with space and altitude on a 0.5° × 0.625° grid and across 35 pressure levels (Fig. S1–S2). The dataset was generated by integrating global model ensembles from 2004–2008 (except WRF-Chem and IMPACT, which covered 2007–2016 and 2004, respectively) with observational data and experimental constraints. The evaluation in Adebiyi et al. (2020) indicated that the mean difference between DustCOMM MEE and observations was ~1%. Compared with the unconstrained model ensemble, DustCOMM better reproduced lower MEE near source regions and higher MEE farther downwind due to transport and aging. For the source and downwind regions examined here

(including TD, TH, TP), several recent studies (Han et al., 2022a, 2022b; Tang et al., 2024), which characterized dust loading and transport based on DustCOMM MEE, have proven the effectiveness of this product to a certain extent. It should be pointed out that although no humidity effect was explicitly accounted in DustCOMM MEE product, some studies suggested that the hygroscopic growth of mineral dust under subsaturated conditions was generally weak, for example the water uptake mass was only ~2% for dry dust at RH = 90% in Qinghai and Xinjiang(Chen et al., 2020), and the diameter growth factor of natural mineral dust was typically < 1.1 for RH ≤ 90% (Koehler et al., 2009). Moreover, we selected the aerosol type of "Dust" rather than "Dust mixture" in CATS. Together with the generally dry background over the study region, we inferred that the humidity-induced enhancement of MEE under this condition was likely on the order of several percent, which was smaller than other uncertainty sources such as extinction retrieval and wind bias.

**3.7.2 Sampling and cloud contamination**

Although CATS provides higher spatiotemporal resolution than some spaceborne lidars such as CALIPSO, its effective observing period spans only about three years and its local-time sampling is not temporally uniform. Therefore, it is important to clarify that the diurnal variations presented in this work do not represent a "true" 3-hourly evolution for any specific day; rather, it depicts the average dependence on local time for different seasons over the study region. Not only aerosol, CATS retrievals also have been used to investigate the diurnal variations of clouds, and planetary boundary layer height (Noël et al., 2018; Yu et al., 2021; Li et al., 2023; Wang et al., 2024). Regarding cloud screening and aerosol typing, the CATS Level-2 product applies a cloud–aerosol discrimination (CAD) algorithm and multiple quality-control tests to separate clouds from aerosols and to flag potential cloud contamination (CATS L2O Profile Products Quality Statements, Version Release 3.00). These procedures include thresholds in cloud fraction, tests based on layer-integrated attenuated backscatter, and relative humidity checks, aiming to better distinguish optically thick clouds from strongly scattering dust/smoke layers and to reduce misclassification (e.g., lofted dust as ice cloud, or dense smoke as cloud). Building on these product-level controls, our processing further applies more strict filtering to improve retrieval reliability (Xiong et al., 2023). Prior studies have also validated the credibility of CATS classification and retrieval results from different perspectives (Proestakis et al., 2019; Nowottnick et al., 2022).

**3.7.3 Extinction wavelength conversion**

A key step of this study was to derive the CATS 532 nm extinction coefficient from CATS 1064 nm measurements, using the extinction conversion ratio of 1064-to-532 nm from CALIPSO observation. Comparisons with CALIPSO Level-3 dust extinction at 532 nm (Fig. S6) show that CATS captured the major spatial patterns and the vertical decay of dust extinction, as well as regional differences, such as the strongest extinction near the surface over the TH with rapid decreases aloft, and an elevated dust layer around ~3 km over the TD. However, relative to CALIPSO, CATS appeared to be slightly higher in the mid-troposphere (~4–6 km). Annual statistics (Fig. S7) further demonstrated overall regional consistency between CATS and CALIPSO, with high correlation (r = 0.88) and small biases (mean bias ≈ 0.01 $km^{-1}$, MAE ≈ 0.01 $km^{-1}$, RMSE ≈ 0.02 $km^{-1}$). These results suggest that CATS retrievals reliably presented the spatiotemporal variations in dust extinction, which ensured the reliability of the analysis on spatial distribution, seasonal variability, and local-time dependence. Following the Ångström relation $\alpha_{550} = \alpha_{532}(\frac{550}{532})^{-\alpha}$ and weak-to-moderate spectral dependence in the visible exhibited by mineral dust,

we adopt α = 0.5–1.0 as the typical range. This choice introduced a mass change of only –1.7 % to –3.3 %. Even under the conservative upper limit of α = 1.5, the shift remains within –5 %. Thus, the mass concentration bias due to the conversion from 532 to 550 nm were only a few percent (<5%).

**3.7.4 Mass concentration, wind, and deposition**

In DMC calculation, the average value of the layer below 180 m is regarded as the near-surface concentration. To evaluate the uncertainty of this approximation, we systematically compared the CATS-derived dust concentrations with reanalysis products (MERRA-2) and surface observations. Seasonal-mean surface dust concentrations between CATS and MERRA-2 showed the correlation coefficients between 0.54 and 0.74. CATS results were slightly higher than MERRA-2 ones in spring (approximately +13 $\mu g \cdot m^{-3}$ of mean bias) but lower more in summer, autumn, and winter (approximately −18 to −27 $\mu g \cdot m^{-3}$), which resulted in an overall underestimation of about 10 to 20 $\mu g \cdot m^{-3}$ in annual means. The ranges of seasonal MAEs (Mean Absolute Error) and RMSEs (Root Mean Square Error) were ~41–68 $\mu g \cdot m^{-3}$ and ~57–98 $\mu g \cdot m^{-3}$, respectively. Compared to the typical concentrations of ~150-250 $\mu g \cdot m^{-3}$ over the study region, the MAE of CATS retrievals was about 20–40% (Fig. S3). Moreover, seasonal-mean comparisons between CATS DMCS and in situ observations of differences between $PM_{10}$ and $PM_{2.5}$ (as a dust proxy) showed correlations of 0.56–0.68, and a slightly positive bias (+8 $\mu g \cdot m^{-3}$) in spring and negative biases (-20 $\mu g \cdot m^{-3}$) in other seasons (Fig. S4). For column DMC, the seasonal correlations between CATS and MERRA-2 were 0.57–0.84, and the seasonal mean bias, MAE, and RMSE were -0.06– +0.01, 0.05–0.09 and 0.06–0.12 $\mu g \cdot m^{-2}$, respectively (Fig. S5). In conclusion, the uncertainty of seasonal-mean DMC retrieval was approximately inferred as 20–40%.

In flux calculation, the bias of wind speed is a major impactor. Some studies indicated that ERA5 10 m wind speed agreed well with surface observations, with a global mean bias of about −4.5% (Fan et al., 2021), and it performs better than MERRA-2 one over East Asia (Li et al., 2025). In addition, the wind speed biases in different vertical layers for ERA5 and MERRA-2 were in a reasonable range (Wu et al., 2024). Given that the regional mean systematic bias is < 5% and the random deviation at different vertical levels is < 0.5 $m \cdot s^{-1}$, we conservatively assume a random uncertainty of 10–20% for ERA5 wind speeds in the subsequent error-propagation analysis. It should be noticed that the boundary fluxes in this work are primarily used to diagnose major transport pathways and their seasonal evolution, rather than to construct a strictly closed mass budget under mass conservation. This study only considered dry deposition and didn't take into account wet deposition, which would lead to a certain degree of overestimation of the flux. However, the precipitation in the study area is relatively small, the major deposition contribution is from dry deposition in most seasons (Fig. S8). Furthermore, when calculating the dust contribution from the Qaidam Basin, this study exclusively accounted for dry deposition while neglecting wet removal processes. This omission introduced a systematic bias that likely inflated the estimated outward dust flux. In brief, considering the combined uncertainties associated with concentration retrieval, wind speed variability, and deposition mechanisms, we estimated an overall error margin of 25–45%. Consequently, these results should be interpreted primarily as a characterization of the spatiotemporal patterns of dust vertical distribution and transport, rather than as a rigorous quantitative budget.